



# Geological-geotechnical analysis of a rock-toppling prone canyon in Furnas, Brazil, after a fatal event

Victor Cabral[1], Fábio Augusto Gomes Vieira Reis[1,2], Joana Paula Sanchez[3]; Rodrigo Irineu Cerri[4]; João Paulo Monticeli[5]; Claudia Vanessa dos Santos Corrêa[1]; Vinícius Queiroz Veloso[1]; Débora Moraes Duarte[6]; Guidotti de Souza dos Garion[6]; George A. Longhitano[6]; Bruno Fructuoso Coelho de Souza[6]; Marcelo Fischer Gramani[7]; Caiubi Emanuel Souza Kuhn[1]; Lucilia do Carmo Giordano[1,2]

[1] Center of Applied Natural Sciences (UNESPetro), Institute of Geosciences and Exact Sciences, São Paulo State University – UNESP. Address: Av. 24A, 1555 – Rio Claro, São Paulo, Brazil.
[2] Department of Environmental Engineering (DEA), Institute of Geosciences and Exact Sciences, São Paulo State University – UNESP. Address: Av. 24A, 1555 – Rio Claro, São Paulo, Brazil.
[3] Faculdade de Ciência e Tecnologia, Universidade Federal de Goiás – UFG. Address: Rua Mucuri S/N – Aparecida de Goiás, Goiás, Brazil.
[4] Department of Geology (DG), Institute of Geosciences and Exact Sciences, São Paulo State University – UNESP. Address: Av. 24A, 1555 – Rio Claro, São Paulo, Brazil.
[5] Geotechnics Department, Engineering School of São Paulo (EESC), University of São Paulo – USP. Address: Av. Trabalhador São-Carlense, 400 – São Carlos, São Paulo, Brazil.
[6] G-Drones. Address: Rua Desembargador do Vale, 653 – São Paulo, São Paulo, Brazil.
[7] Institute for Technological Research – IPT. Address: Av. Prof. Almeida Prado, 532 – São Paulo, São Paulo, Brazil.

*Correspondence to*: Victor Cabral (victor.carvalho@unesp.br)

**Abstract.** When a disaster related to a natural phenomenon occurs in areas dependent on Geotourism, restoring tourist confidence can be a challenge. An example is the rock-toppling event that occurred on January 2022 in one of the four canyons located in the Furnas reservoir in Brazil, which caused 10 fatalities. Visitation fear and their temporary closure severely impacted the economy of the surrounding municipalities that rely on tourism. To support a safer operation of the canyons to visitation, our study investigates the factors that can lead to landslide events in the region, based on the combination of field investigations, rock-mass quality evaluation (RMR14) and kinematic analysis. We hypothesize that the assessment of rock-mass quality can successfully identify specific areas in the bedrock that are prone to rockfall and rock toppling, supporting risk management strategies in tourist regions. Our results indicate that the 2022 rock-toppling event occurred due to the combination of different factors, such as rainfall infiltration in the unfavorably-oriented joints of the bedrock and reservoir water-level fluctuations. Moreover, the long-term erosion at the base of the slope, caused by the nearby waterfall flow and water-level variations, weakened rock-mass support. The RMR14 method adapted to open rock slopes successfully supports the estimation of the bedrock's geomechanical properties, identifying structural zones in the rock mass that are prone to slope failure(s) and, as a consequence, should be monitored. The kinematic analysis further indicates that the four canyons are highly susceptible to planar failures and, less so, to toppling, although specific locations in the slopes show a higher rock-toppling susceptibility, especially where two perpendicularly-oriented fault zones (NW-SE and NE-SW strikes) intersect. The consideration of geomechanical properties in hazard evaluation is recommended as a risk management strategy, supporting the delimitation of regions near the rock slopes that should be restricted and of specific portions in the bedrock that should be retained. Our study was fundamental to establishing visitation procedures in the canyons, so that tourists and workers are more protected and aware of the existing geohazards.

**Keywords:** Landslides, Structural geology, Rock-mass quality, Kinematic analysis, Natural hazards.



**Graphical abstract**

## STEPS THAT LED TO THE ROCK-TOPPLING EVENT

**1. Erosion**

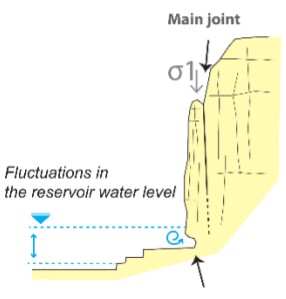

Scouring of the base of the slope, due to the nearby waterfall flow and, potentially, the reservoir water-level fluctuations

**2. Tilting of the block**

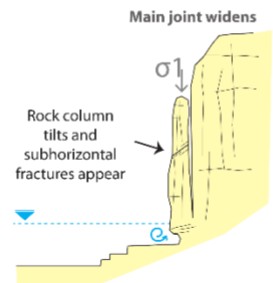

**3. Slope failure**

Largely-spaced joints contribute to increasing pressure on the rock mass. The triggering mechanism is rainfall, with the percolation and accumulation of water in the joints triggering the slope failure

**4. Topple**

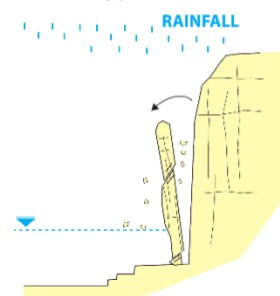

Rotation of the rock column, with partial disaggregation

**5. Breakdown**

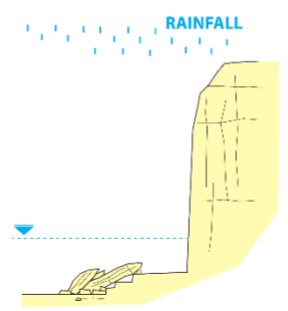

Total disaggregation of the rock block due to the impact with water and bedrock surface



## 1 Introduction

Tourism in natural areas, commonly referred to as Geotourism, has grown extensively worldwide in the last decades (Dowling 2013; Dowling and Newsome 2017). Geotourism is a form of tourism that is mainly focused on the landscape, whether natural or modified by humans (Ólafsdóttir and Dowling, 2014; Kuhn et al., 2022), and its promotion can support a more sustainable tourism practice (Dowling and Newsome, 2017; Ólafsdóttir and Tverijonaite, 2018). Geotourism can also be economically positive for a region or country (Dowling and Newsome, 2017), with the potential of supporting the natural environment

preservation. While the COVID-19 pandemic caused a sudden halt in tourism worldwide (Yang et al., 2021), a quick recovery of the sector is occurring since 2023 (UNWTO, 2024).

Disasters triggered by natural phenomena can cause profound impacts on individuals, communities and, consequently, on tourism (Ruan et al., 2017), as they are a product of the increasing complexity of social-ecological systems (Becken and Hughey, 2013). With the expected increase in the frequency of high-intensity to extreme rain events in Brazil (Pereira Filho et

al 2014; Marengo et al. 2021; Lopez et al. 2023; Marengo et al. 2024), more frequent disasters related to hydrogeomorphic processes are also expected, such as floods and landslides. These processes are the responsible for the highest socioeconomic impacts related to natural hazards in the country (Alvalá et al., 2019; Cabral et al., 2022), affecting particularly municipalities that depend on Geotourism (Rocha and Matedi, 2017).

The 2022 fatal rock-toppling event in the region of the Furnas canyons (Minas Gerais state, Southeast Brazil) is a recent

example. In this event, a rock column toppled on top of two boats, causing 10 fatalities, which led to the closure of all the four canyons in the region and significantly affected the tourism-dependent municipalities nearby. As Berdychevsky and Gibson (2015) point out, disasters with natural causes not only have the potential to affect the tourism industry, but can also threaten the socioeconomic foundations of municipalities, states and even countries (Liu, 2014; Ruan et al., 2017). Understanding, managing and responding to disaster hazard and risk, therefore, should be an integral component of Geotourism (Wilks and

Moore, 2004; Becken and Hughey, 2013; Shakeela and Becken, 2015).

In geohazard studies that involve rock slope stability, textural (2D) and structural (3D) data are fundamental to characterizing rock-mass structure and quality, as the occurrence of discontinuities and anisotropies in the rock body is directly related to their strength, deformability potential and long-term equilibrium (Jaboyedoff et al., 2004; Bajni et al., 2022). Several methods have been proposed to assess rock-mass structure and quality (e.g., Rock Quality Designation index - RQD, Deere et al., 1967;

"Q" method, Barton et al., 1974), with the two more widely used in geomechanics arguably being the "Rock Mass Rating" (RMR) (Bieniawski, 1976; 1989) and the "Geological Strength Index" (GSI) (Hoek and Brown, 1997). More recently, the RMR method was updated by Celada et al. (2014) in the RMR14, based on 25 years of experience using the classification method.

While widely applied in the field of mining and engineering (Zhang et al., 2019), the analysis of rock-mass quality has also

been employed in several open-slope studies, especially the adaptation "Slope Mass Rating" (SMR) proposed by Romana (1985; 1993) based on the original RMR method by Bieniawski (1976; 1989). The SMR has been applied especially in the

analysis of road cuts (Siddique et al., 2015; Chaurasia, 2017), and its main advantage in relation to the traditional RMR is the consideration of the relationship between the relative orientation of the rock slope and its discontinuities (Irigaray et al., 2003). In this context, we hypothesize that the updated RMR14 can also be adapted to open-slope studies focusing on geohazards, supporting hazard and risk management strategies in the identification of areas in the bedrock that are more prone to rockfall and rock-toppling. Our objective is to analyze the factors that can lead to landslides in the Furnas canyons, supporting the establishment of procedures for a safer operation of the area, so that visitors and workers are more protected and aware of the existing geohazards. Our investigation is based on extensive field campaigns and aerial surveys, which supported both the structural geology and kinematic analysis, as well as the rock-mass quality evaluation.

## 2 The Furnas canyons and the rock-toppling event of January 2022

Rock toppling and rockfall are landform processes that naturally occur in the four canyons located in the region of Furnas (Minas Gerais state), denominated (1) Capitólio (where the fatal event occurred), (2) Cascatinha, (3) Tucanos and (4) Cabritos canyon (**Figure 1**). The "Furnas canyons" are a very popular tourist attraction in Brazil, as they are located in one of the largest reservoirs in the country (Machado et al., 2020), with an area of over 1,440 km$^2$. The reservoir was created due to the construction of the Furnas Hydropower plant in 1958 (Godoy, 2017), and, in 2020, the tourism related to the canyons and the reservoir represented over 65% of the total Gross Domestic Product (GDP) of the surrounding municipalities, with the population tripling during holidays (Machado et al. 2020).

Overtourism was a common practice prior to the event, resulting in a lack of control of tourist safety, as well as in a deficiency or even absence of environmentally-sustainable practices (Vieira et al., 2022). Overtourism can potentially be one of the contributors to the occurrence of the fatal disaster of January 2022, as there was no control in the numbers of people/boats allowed at the same time in the canyons, as well as no safety distance from the canyon walls established. **Figure 2** shows the canyon before and after the event, where the location of the toppled rock column can be observed due to a less weathered portion in the bedrock.

The fatal rock-toppling event occurred on January 8, 2022, and was potentially triggered by a 50 mm rainfall accumulated in 96 h, which cannot be considered particularly intense. However, when the rainfall one-month prior to the collapse of the 500-ton rock column is considered, the accumulated precipitation reaches approximately 355 mm, higher than what is expected for



summer months (ca. 280 mm). It is important to note that the lack of a strong pluviometer net in the region can underestimate

rainfall indices in the canyons.

**Figure 1: The four canyons located in the Furnas reservoir, state of Minas Gerais, Southeast Brazil. The fatal rock-toppling event occurred in the Capitólio canyon (1). The Furnas Reservoir was created due to the construction of the Furnas hydropower plant. The source of the aerial image is from Google Maps ©.**

The geology is comprised of metasediments of the Araxá Group, mainly quartzite, schist and gneiss (Heilbron et al., 2007).

Fine-grained quartzite predominates in the canyons, denominated "Furnas Quartzite", with a strong sub-horizontal foliation

(Heilbron et al., 2007). The tectonic context of the Araxá Group is the Passos *Nappe*, with four phases of deformation (Heilbron

et al., 2007), which resulted in a heavily jointed and fractured rock mass, prone to landslides.



The Furnas Quartzite has a thickness that varies from 30 to 100 m in the study site, interbedded with centimetric to decimetric beds of muscovite schist and quartz-muscovite schist. According to Heilbron et al. (2007), the four phases of deformation can be summarized in: (i) $D_1$ and $D_2$ – these two phases occurred under high temperatures (400 – 750ºC), with the formation of

recumbent folds and two foliation sets ($S_1$ and $S_2$) parallel to the bedding ($S_0$); (ii) $D_3$ and $D_4$ – these two phases occurred under lower temperatures (< 400º C), forming gentle/open folds ($D_3$), with steep axial surfaces and axes with a NW-SE strike. The $D_4$ deformation also formed open to gentle folds with N10W-N20E axes and the faults parallel to the axial surface of these folds are frequent in both phases, with the interference of $D_3$ and $D_4$ folds typically forming the pattern of "domes and basins".

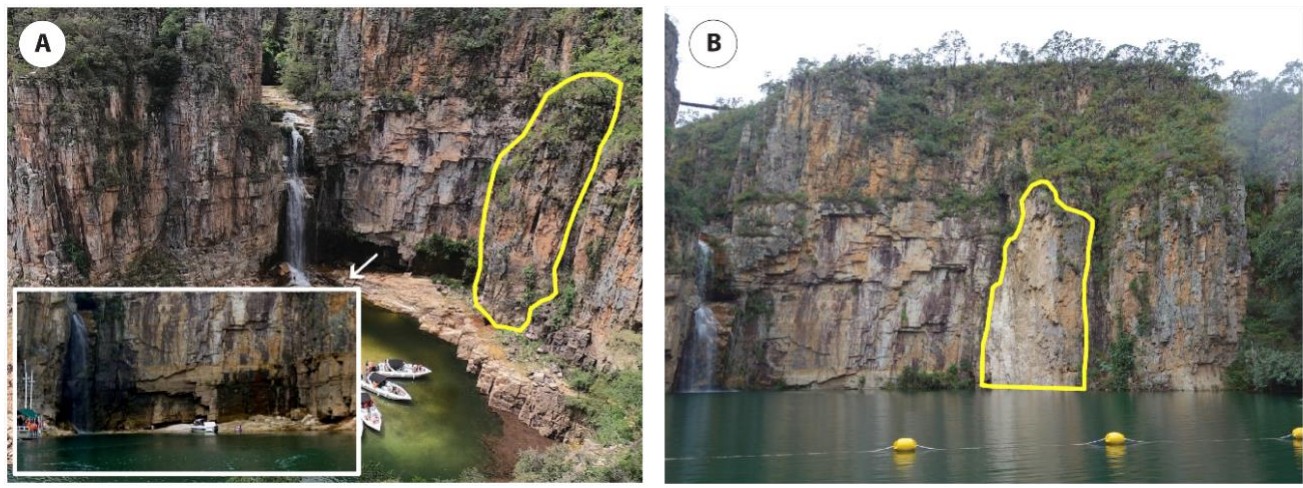

**Figure 2: Overview of the Capitólio canyon before (A) and after (B) the fatal event. A) The large rock column that toppled is highlighted in yellow. Note the intense scouring at the base of the canyon wall (Source: Ion David Zarantonelli, 2019). The agglomeration of boats in the canyons was a common practice before the catastrophe. B) The location of the rock column that collapsed is indicated in yellow, highlighted by a less weathered (lighter) portion in the bedrock. Reservoir water level was much higher in March 2022 (B) than in July 2019 (A).**

**3 Methods**

For the analysis of the factors that can lead to slope failure(s) in the canyons, we first investigated, *in situ*, the geological and geotechnical characteristics of the bedrock. Then, high-resolution photogrammetric products (aerial photographs, orthophotos, 3D models) were acquired, which supported the structural analysis, the rock-mass quality assessment and the kinematic analysis.

**3.1 Field investigation and data collection**

Two field campaigns were conducted in the canyons (February and March, 2022), in which an overview of the geology, geomorphology and rock-mass conditions was performed (**Figure 3A**). Structural measurements were also taken, as well as the physical properties of the bedrock were estimated using the Schmidt Hammer (**Figure 3B**).



The structural geology analysis was conducted with a Clar-type geological compass, where measurements of the dip and dip direction of planar (fault, joint and foliation) and linear (fold axis, lineament) structures were taken. In total, 557 total measurements were made in the study site and the collected data was processed using the Stereonet® software (Allmendinger et al., 2017), enabling a clearer visualization and analysis.

The *in-situ* bedrock-strength data was measured using the Schmidt Hammer (L-type, from the maker Proceq®), which was fundamental in the estimation of the geomechanical properties of the bedrock and its discontinuities, such as the uniaxial compression strength (UCS) and weathering degree, which are applied in the RMR14 method. The test procedure of the American Society for Testing and Material (ASTM) was followed in the Schmidt Hammer data collection.

The estimation of the UCS based on the Schmidt Hammer data follows Barton et al. (1974) and Barton and Choubey (1977), which established an empirical method to express the shear strength of the discontinuity planes in rock masses (Eq. 1). According to Barton and Choubey (1997), for unweathered discontinuity planes, the basic friction angle ($\phi_b$) is applied in Equation 1 and the JCS (joint wall compression strength) is equal to the UCS. However, as the discontinuity planes often show different weathering degrees, especially in tropical areas, the JCS will be lower than the UCS.

To address the weathering effect, instead of the basic friction angle ($\phi_b$) the residual friction angle ($\phi_r$), calculated according to Equation 2, is applied in Equation 1. Both the basic friction angle ($\phi_b$) and JCS are estimated according to Schmidt Hammer data. For a more detailed characterization of the estimation of geomechanical properties based on the Schmidt Hammer, we refer Barton and Choubey (1977).

$$\tau = \sigma_n \cdot tg\left[\left(\phi_b \; or \; \phi_r\right) + JRC \cdot log_{10}\left(\frac{JCS}{\sigma_n}\right)\right] \tag{1}$$

Where $\tau$ = Shear strength; $\phi_r$ = Residual friction angle of discontinuity; $\sigma_n$ = Normal stress on the discontinuity plane; JRC = Discontinuity roughness coefficient; and JCS = Strength of the discontinuity surface.

$$\phi_r = (\phi_b - 20) + 20\left(\frac{r}{R}\right) \tag{2}$$

Where $\phi_b$ = Basic friction angle of discontinuity; R = Schmidt rebound hammer hardness value of the fresh surface; and r = Schmidt rebound hammer hardness value of the weathered surface.

Water level data (elevation) was retrieved from Brazil's Water Agency (ANA – *Agência Nacional de Águas*), through the Reservoir Monitoring System (SAR – *Sistema de Acompanhamento de Reservatórios*), available at: https://www.ana.gov.br/sar0/MedicaoSin?dropDownListEstados=14&dropDownListReservatorios=19004&dataInicial=01%2F01%2F1990&dataFinal=15%2F04%2F2024&button=Buscar. At the time of our investigation (March 2022), the reservoir water level was at about 765 m above sea level (asl).

Rainfall data was retrieved from different sources. Monthly rainfall data from 1990 to 2012 was acquired from the meteorological database of the National Institute of Meteorology (INMET – *Instituto Nacional de Meteorologia*), based on the pluviometer "Furnas Hydropower plant (UHE Furnas)". As this pluviometer was inactive after 2012, rainfall data from 2013 to January 2022 was retrieved from the pluviometer "Boa Esperança", the nearest one upstream the canyons. The data from





the "Boa Esperança" pluviometer was obtained from the National Center of Monitoring and Early Warning of Natural Disasters

(CEMADEN - Centro Nacional de Monitoramento e Alerta de Desastres Naturais).

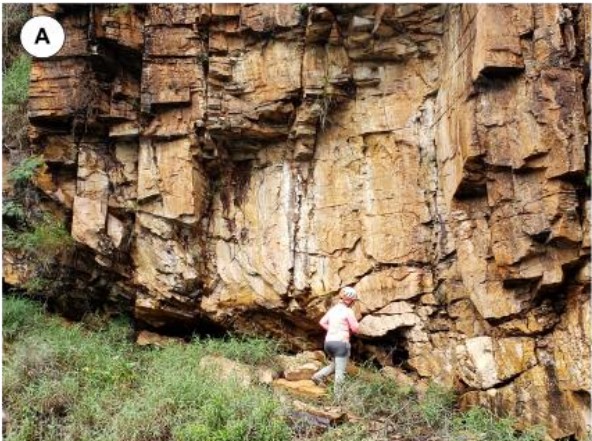
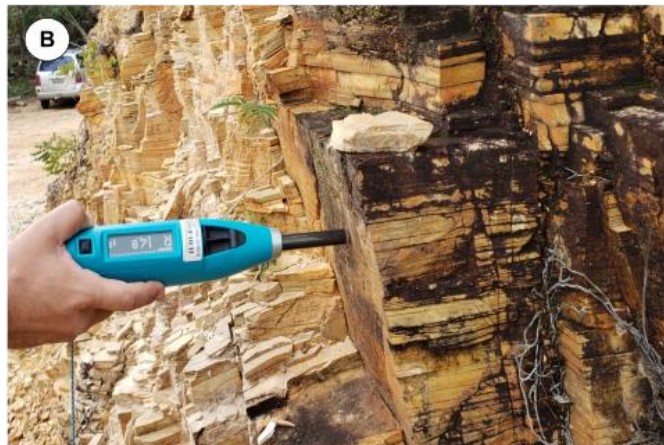

**Figure 3: Field investigations. A) Structural data measurements. B) Schimdt Hammer sampling.**

Finally, informal interviews were conducted with the surrounding landowners, boat sailors, as well as the engineering team of the municipalities that comprise the four canyons, to better understand how tourist activities were conducted prior to the

disaster, as well as to inquire about variations in reservoir water level and records of past landslides. Despite an official request made by the municipality of Capitólio for the company that controls the hydropower plant to provide geotechnical monitoring data of the reservoir/canyons, no data was provided to our team of researchers, which was the only one authorized by the municipality to conduct post-disaster assessments.

### 3.2 Acquisition of photogrammetric products

The acquisition of photogrammetric products was fundamental for our study, as no high-resolution data (e.g., topography, ortophotos) was available for the region. The primary data from the Unmanned Aerial Vehicle (UAV) was acquired using the device Phantom Pro 4 (DJI), equipped with a RGB camera, model FC3170 and a sensor of 1'' CMOS for the imagery acquisition. The camera has a focal length of 4 mm and acquired images with a 20-megapixel resolution in the visible wavelengths (Red, Green and Blue). The resulted products have a size of 5472 x 3648 pixels and aspect ratio of 3:2, with the

Ground Sampling Distance (GSD) of 1 cm. For the orthomosaic, the flying altitude was of 150 m, with a field of view of 80°. Forward and lateral overlap are of 80% and 70%, respectively. The flights were conducted during 4 days in May, 2022, and a total of 28h of flights was necessary for the UAV imagery acquisition.

The processing of the photogrammetric products was made using the software *Agisoft Metashape*. The total area covered by the UAV was approximately 1,836,943 m²: 580,102 m² in the Capitólio canyon, 531,151 m² in the Capivara canyon, 213,323

m² in the Tucanos canyon and 512,367 m² in the Cabritos canyon. The images collected with UAV were processed into orthophotos and grid-based DSMs (Digital Surface Models), using a Structure from Motion (SfM) workflow (Lowe, 2004).



The use of drone navigation GNSS (±3cm) allowed the georeferencing in a coordinate system, which was exported to a high-resolution (1.65cm/pixel) grid-based DSM and Orthomosaics. The algorithms implemented in the software are described in Verhoeven (2011).

The main products that resulted from the UAV survey were the Digital Surface Model - DSM, the 3D model and the orthophotos of the four canyons. We chose to employ the DSM instead of DTM (Digital Terrain Model) due to uncertainties when generating the latter based on the UAV survey. Moreover, since no infrastructures were installed in the canyons and as the vegetation on top of the slopes consist mainly of shrubs of up to 1 m (Cerrado Biome), they did not considerably impact the rock-slope height estimation. The 3D model of the canyons (**Figure 4**) allowed a detailed reconstruction of the slope faces,

supporting *in-situ* observations.



**Figure 4: The 3D model of the Capitólio canyon.**

**3.3 Rock-Mass Quality Assessment**

The RMR14 methodology, proposed by Celada et al. (2014), was adopted and adapted to assess rock-mass quality in the

canyons. The RMR14 maintains three parameters of the original RMR methodology (Bieniawski, 1989): (1) uniaxial



compression strength of the rock (UCS); (2) number of discontinuities per meter; and (3) water effect. The RMR14 proposes a revision on the last two parameters of the original RMR, related to the (4) condition of the joints in the rock mass and to the (5) intact rock alterability due to water (swelling). These five parameters are denominated $RMR_b$, or RMR basic, as the effect of excavation is not considered (Celada et al., 2014).

Besides proposing a revision on some of the RMR parameters, RMR14 adds three adjustments factors, related to (1) the orientation of the tunnel axis in relation to the main set of discontinuities in the rock mass (always negative, $F_0$), (2) the excavation method ($F_e$), and (3) the stress-strain behavior of the rock mass at the tunnel face ($F_s$) (Eq. 3). As in our case, there is no tunneling or excavation method involved, parameters $F_e$ and $F_s$ are disregarded (Eq. 4).

$$RMR14 = (RMR_b + F_0) * F_e * F_s \qquad (3)$$

$$RMR14 = RMR_b + F_0 \qquad (4)$$

The parameter $F_0$ is adapted to consider the orientation of the canyon wall instead of the tunnel axis, as the dip and dip direction

of the discontinuities in relation to the canyon wall orientation are believed to play an important role in slope failure initiation. The adaptation of the $F_0$ follows what is proposed in the SMR method (Romana, 1985;1993), being a function (Eq. 5) of the parallelism between the strike of the joints and the slope face ($F_1$), the dip of the joints ($F_2$) and the relationship between joints dip and slope dip ($F_3$). Since we are dealing with a natural slope, the adjustment factor ($F_4$) has a rating of +15. **Table 1** shows the scores of each adjustment factors, as well as how to analyze their relationship.

$$F_0 = (F_1 * F_2 * F_3) + F_4 \qquad (5)$$

**Table 1: Relationship between the canyon walls and structural geology, which can impact slope stability. Based on Romana (1993). αj = joint strike; αs = slope face strike; βj = joint dip; βs = slope face dip.**

| Relationship between slope face and structural geology | Very favorable | Favorable | Fair | Unfavorable | Very unfavorable |
|---|---|---|---|---|---|
| F1 = \| (αj - αs) - 180°\| | > 30° | 30° - 20° | 20° - 10° | 10° - 5° | < 5° |
| *Score* | 0.15 | 0.4 | 0.7 | 0.85 | 1.00 |
| F2 | | | - | | |
| *Score* | | | 1.00 | | |
| F3 = βj + βs | <110° | 110° - 120° | 120° - 150° | N/A | N/A |
| *Score* | 0 | -6 | -25 | N/A | N/A |
| F4 | | | Natural Slope | | |
| *Score* | | | +15 | | |

The rating of the first parameter of the $RMR_b$, UCS, has a maximum score of 15 points (**Figure 5A**) and it was estimated using data from the Schmidt hammer. The second parameter, the number of discontinuities per meter, has a maximum score of 40

points (**Figure 5B**) and was estimated based on field observations and on the 3D model of the canyons, in which detailed





information about the rock mass was able to be interpreted (especially in the upper sections of the canyon wall). The third parameter, water effect, has a maximum score of 15 points (**Table 2**) and was also based on visual interpretation of the 3D model of the canyons, as well as on our field observations.

**Table 2: Rating of each parameter related to the presence of water in the discontinuities of the rock mass (Celada et al., 2014).**

| **Ground State** | Dry | Slightly humid | Humid | Dripping | Water flow |
|---|---|---|---|---|---|
| **Score** | 15 | 10 | 7 | 4 | 0 |

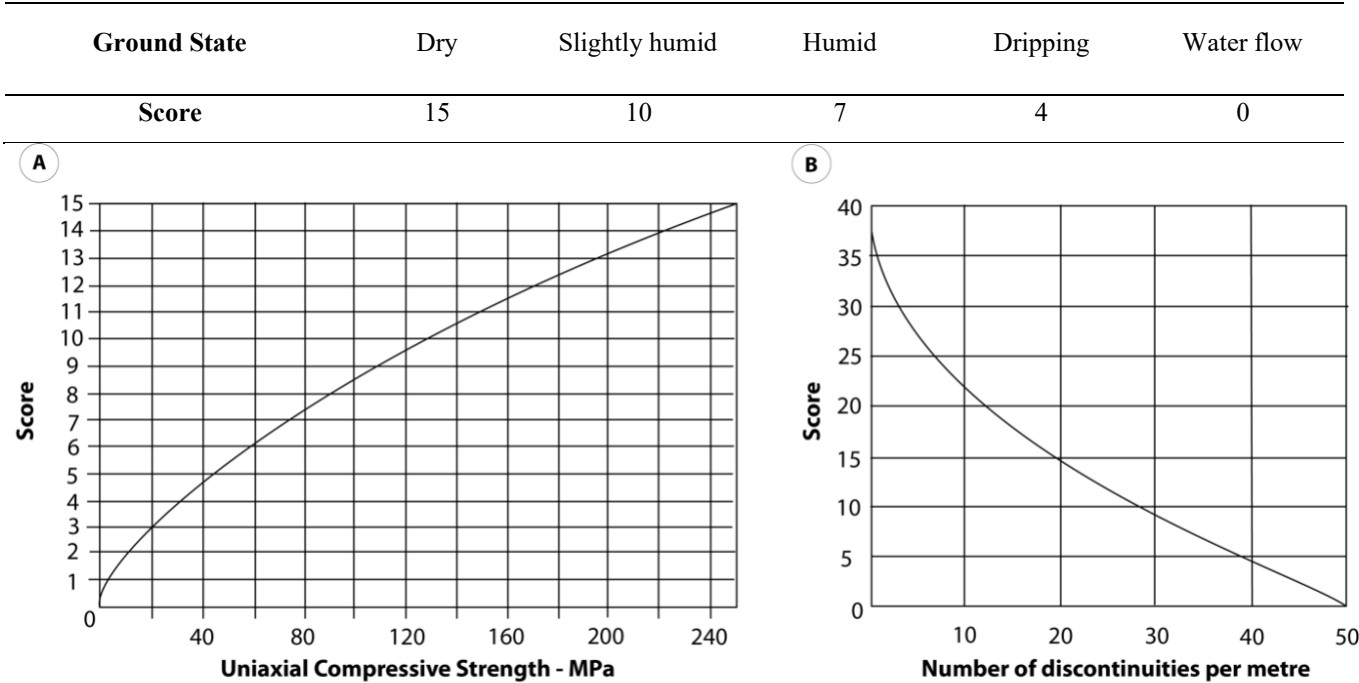

**Figure 5: Scores (or rating) for the uniaxial compressive strength of the rock mass (A) and the number of discontinuities per meter (B). Adapted from Celada et al. (2014).**

The analysis of the joint condition has a maximum score of 20 points and is based on four aspects: (i) persistence of discontinuities, observed during our field investigations; (ii) roughness of the discontinuities measured through the Joint Roughness Coefficient (JRC), estimated using the Schmidt hammer; (iii) Infilling type in the discontinuities, observed *in situ* during field campaigns; and (iv) weathering degree of the discontinuity planes (**Table 3**), also estimated *in situ* and based on Schmidt Hammer data. The intact rock alterability has a maximum score of 10 points (**Table 3**) and is based on aerial photographs and the 3D model.

The final parameter considered in our study that is part of the RMR14 method is the orientation of the discontinuities in relation to the orientation of the canyon walls. The orientation of the discontinuities was measured in the field using a Clar-type compass and compared to that of the canyon walls (**Table 1**).

The total sum of the scores attributed to the six parameters considered in our study are finally calculated, with the result representing the rock mass quality index or Rock Mass Rating (RMR14) (**Table 4**).





**Table 3: Rating of each parameter related to the condition of discontinuities in the rock mass and the intact rock alterability (Celada et al., 2014).**

| Discontinuities condition | | | | |
|---|---|---|---|---|
| *Persistence* | < 1 m | 1 - 3 m | 3 - 10 m | > 10 m |
| | 5 | 4 | 2 | 0 |
| *Roughness* | Very rough | Rough | Smooth | Slickensided |
| | 5 | 3 | 1 | 0 |
| *Gouge infilling* | Hard | | Soft | |
| | < 5 mm | > 5 mm | < 5 mm | > 5 mm |
| | 5 | 2 | 2 | 0 |
| *Weathering* | Unweathered | Moderately weathered | Highly weathered | Decomposed |
| | 5 | 3 | 1 | 0 |
| Intact rock alterability | | | | |
| Alterability $I_{d2}$ (%) | | | | |
| | > 85 | 60 - 85 | 30 - 60 | < 30 |
| | 10 | 8 | 4 | 0 |

**Table 4: Ratings of the RMR14 methodology, considering rock mass quality (Celada et al., 2014).**

| Final rating | 100 - 81 | 80 - 61 | 60 - 41 | 40 - 21 | < 21 |
|---|---|---|---|---|---|
| **Class** | I | II | III | IV | V |
| **Description of the rock mass** | Very good | Good | Normal | Weak | Very weak |
| **Cohesion of the rock mass (kPa)** | > 400 | 300 - 400 | 200 - 300 | 100 - 200 | < 100 |
| **Internal friction angle of the rock mass (º)** | > 45 | 35 - 45 | 25 - 35 | 15 - 25 | < 15 |

### 3.4 Kinematic analysis

Kinematic analysis determines the potential failure modes that can occur in a jointed rock mass, based on the slope geometry, material properties and angular relations between slope face and bedrock discontinuities (Hoek and Bray, 1981). In our study,



Rocscience DIPS (Rocscience, 2024) is applied in the kinematic analysis, which was conducted in all the walls of the four canyons, considering planar, wedge and toppling failure.

For a planar failure, the following geometrical conditions should occur, according to Norrish and Wyllie (1996): (i) the dip
direction of the discontinuity must be within ± 20 degrees of the dip direction of the slope face; (ii) the dip of discontinuity ($\Psi$p) must be less than the dip of slope face ($\Psi$f) and (iii) the dip of discontinuity must be greater than the friction angle of failure plane ($\phi$).

For a wedge failure, the geometrical conditions required are: (i) the azimuth of line of intersection must be similar to the dip direction of the slope face; (ii) the plunge of line of intersection must be less than the dip of slope face; (iii) the plunge of line
of intersection must be greater than the friction angle of failure plane (Norrish and Wyllie, 1996).

For a toppling failure, the necessary conditions are: (i) the difference between the strikes of discontinuity and slope face must be +30° or -30°; the pole of the discontinuity must plot the critical area; and (iii) the following equation must be provided (Eq. 6):

$$\left(90 - \psi_p\right) \leq \psi_f + \phi \qquad\qquad (6)$$

The friction angle of the discontinuities ($\phi$) was based on Schmidt Hammer data, considering both unweathered bedrock, where
peak friction angle ($\phi_p$) was applied, and highly weathered bedrock, where residual friction angle ($\phi_r$) was applied. The consideration of different weathering degrees supports a more detailed kinematic evaluation of the canyons.

## 4. Results

The factors that can lead to rock-toppling events in the canyons are first presented and, then, we analyze how rock-mass quality evaluation can support the identification of areas that are more prone to landslides.

### 4.1 Rock-toppling dynamics and the structural geology of the canyons

The field surveys revealed that the long-term erosion at the base of the rock slope, caused by the adjacent waterfall flow and, potentially, the fluctuations in the reservoir water level, was one of the main causes of the January 2022 rock-toppling event (**Figure 6**). The erosion at the bottom portion of the slope caused the middle section to lose underlying support, with an added pressure of the water accumulated in the joints. As pointed by Cruden (1991), self-weight alone is not sufficient to cause
toppling, with external forces needed to initiate the process.

Besides the erosion of the canyon wall's bottom portion (**Figure 7A** and **7B**), the pervasive sets of unfavorably oriented joints (discontinuities) also strongly contributed to the slope failure. The joint sets often showed very large aperture (up to 50 cm), with the discontinuity planes exhibiting highly weathered surfaces with the formation of soil and vegetation (**Figure 7C**), especially at the top portion of the canyon wall. The infiltration of water in the bedrock's joint system potentially contributed
with the "added pressure" needed for the initiation of the slope failure.





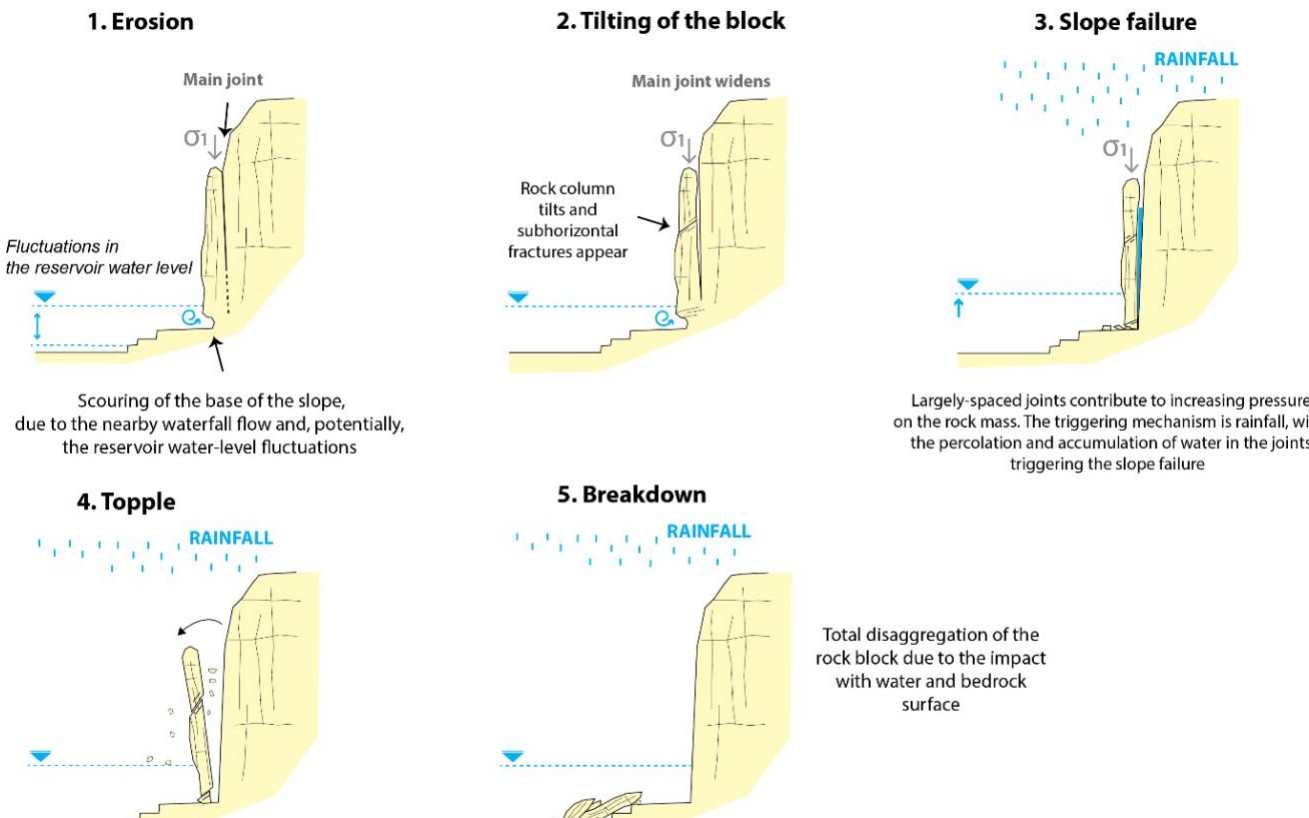

**Figure 6: Steps that led to the catastrophic rock-toppling event in the Capitólio canyon, with $\sigma_1$ representing the main stress tensor. 1) The intense scouring at the base of the slope, due to the waterfall flow nearby and fluctuations in the reservoir water level, undermined the underlying support of the heavily-jointed rock column. 2) Due to weathering, the sub-vertical joints widen and sub-horizontal fractures appear due to its own weight in middle and bottom portions of the rock column. 3) Water percolates and accumulates in the joints and fractures, increasing pressure on the rock mass, leading to slope failure. The increase in reservoir water level raises the local groundwater level and, consequently, the stress conditions in the rock mass, contributing to the occurrence of instabilities. 4) The toppling event occurs and (5) the rock column disaggregates, due to the impact with water and the bedrock surface.**

Moreover, water-level fluctuations in the reservoir may have also contributed to the slope failure. The annual variations of the Furnas water level in the last 32 years (**Figure 8A**) show that after a sharp decrease in 2013, the reservoir level has been variating greatly, contrasting with the long period of somewhat stable levels between 2001 and 2012. The sharp decrease in 2013 can be associated with the drought that Southeast Brazil experienced in the 2013 – 2017 period (Finke et al., 2020), as reservoirs can be significantly affected by changes in annual and seasonal precipitation, as well as temperature variations

(Mukheibir, 2013).



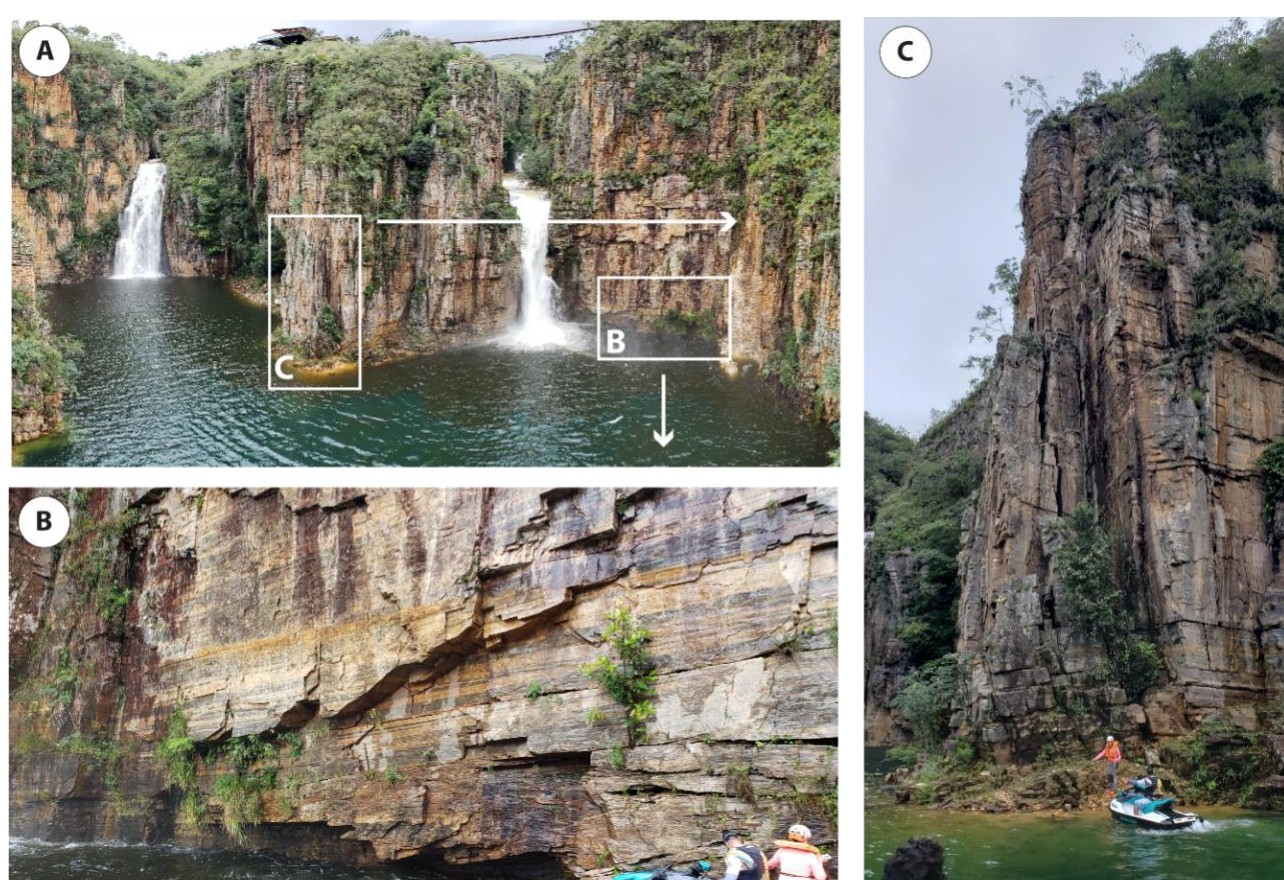

**Figure 7: Details of the Capitólio canyon bedrock. A) Overview of the central area of the Capitólio canyon, showing where photos B) and C) were taken. B) A "cave" formed at the bottom of the canyon wall, due to the scouring of the rock mass due to the nearby waterfall flow and variations in the reservoir water level. C) The pervasive set of joints in the bedrock, with large aperture (up to 50 cm), is clearly seen in this portion. In this particular area, two fault zones intersect (NW-SE and NE-SW strikes), contributing to the increase in jointing and fracturing of the rock mass and facilitating slope failures. In this portion, the isolation of the area or the implementation of a retention structure is suggested, as there is a high susceptibility to toppling.**

When the water level in the month leading to the slope failure is considered (**Figure 8B**), we can observe that there was a 3 m increase in this period, from 755.8 m asl on December 8, 2021, to 758.87 m asl on January 8, 2022. In the seven-days period prior to the catastrophe, the average water-level increase rate was 20 cm/day. Rapid increases in water level and reservoir volume can profoundly impact the local groundwater level and stress behavior on the slope, which, as highlight by Jin et al. (2023), can cause deformation and slope failure(s). The rock-toppling event in the Furnas reservoir, therefore, can potentially be attributed both to the rapid water-level increase and rainfall infiltration in the joint system, combined with the weakened support of toe of the slope due to erosion.



**A** 32-years record of rainfall and reservoir water level fluctuations

**B** Record of daily rainfall and reservoir water level fluctuations leading to the rock-toppling event

**Legend:** ▮ Rainfall ▬ Reservoir water level

**Figure 8: Water level of the Furnas reservoir and rainfall record. A) Long-term (32 years) variations in reservoir water level (red line) and rainfall data (blue bars). B) Reservoir water level variation in the month leading to the rock-toppling event.**

Even though, as highlighted by Nakamura and Wang (1990), most landslides in reservoirs occur during the drawdown or impoundment, more recent studies have shown that water-level fluctuations have also the ability to significantly impact slope stability, creating a stronger water conductivity in the upper portions of the bedrock in periods of higher water level, thus affecting deformation and tensile strength (Tu et al., 2020; Li et al., 2024). A very fractured bedrock, such as the case in the Furnas canyons, can potentially be more affected by such effects.

The structural geology analysis of the canyons further allowed a more detailed assessment of the influence of discontinuities on the toppling event. In the Furnas Quartzite, three family of joints and two perpendicular fault zones were identified, which include:

- Family $F_1$: sub-horizontal joints, often associated with the bedrock foliation, which preferential strike is N75W/15 NE (**Figure 9A**). The thin layers of more micaceous facies interbedded with the quartzite control the sub-horizontal fracture planes;





- Family F₂: NW-SE strike, with dip that varies from 80º to 90º, with a primary SW dip direction and, secondarily, NE dip direction (**Figure 9A**);

- Family F₃: NE-SW strike, dip direction SE (primary) and NW (secondary), with subvertical dip (80º – 90º) (**Figure 9A**);

- Family F₄: Unique structures associated with NE-SW geological faults, related to the decrease in joint spacing and increase in joint frequency, conditioning the shape of the canyons (**Figure 9B**).

**Figure 9. Structural analysis. A) Structural analysis of the bedrock using Stereonet. Three main joint sets were identified in the study (n = 577). B) Overview of the Capitólio canyon, highlighting areas where two fault zones intersect. The rock-toppling event occurred in the far right (to the east) intersection, highlighted in the dotted circle. C) The fracture density in the bedrock is intensified in areas where two perpendicular fault zones intersect. The photograph is from the portion where the fatal rock-toppling event occurred, highlighting the contribution of fractures and joints to the slope failure.**




The rock mass is conditioned by NW-SE and NE-SW subvertical joint sets and by sub-horizontal joints associated with the foliation and bedding. These three different families ($F_1$, $F_2$ and $F_3$) control the formation of regular blocks in the rock mass, in a parallelepiped-like shape. Moreover, singular structures, such as geological faults, shape the canyons, as well as erosive processes at the base of the slopes caused by the action of waves resulted from waterfalls and boats, in addition to the

fluctuations in reservoir water levels. It is observed that these NE-SW faults condition the geometry of the entrance of the canyon, as well as the lateral extremities on the left and right side, forming rectilinear rock walls (**Figure 9B**). The same is observed for the NW-SE joint sets.

Near the intersection of these perpendicular-oriented fault planes, the density of fractures in the bedrock is higher, showing a stronger persistence in the canyon walls, extending from the base to the top of the rock slope (**Figure 9C**). The rock column

that toppled in the event of January 2022 was located in the intersection of these two fault zones (**Figure 9B and 9C**), with a clear increase in discontinuities density.

## 4.2. Rock-mass quality evaluation and the kinematic analysis

In total, 33 measurements with the Schmidt hammer were sampled in the canyons, which supported the overall estimation of the uniaxial compressive strength (UCS) of the rock mass, the compressive strength of the joint walls (JCS) and the residual

and peak friction angles (**Table 5** and **Table 6**), which are all applied in rock-mass quality analysis.

The average rebound values (Q) estimated with the Schmidt hammer varied from 83.3 to 3.1, indicating a wide range in the physical-mechanical response of the rock mass (from very high to low strength). This variation is directly related to different weathering degrees of the bedrock (**Figure 10**), which shows portions of intact rock and very weathered/saprolite portions.

The strength results of the rock mass (**Table 6**), while estimates of real parameters, were fundamental in the geomechanical

classification using the RMR14. In the geomechanical classification of the Capitólio canyon, for instance, ten compartments were individualized (**Figure 11A**), representing each canyon wall. Each compartment was, then, subdivided in smaller structural zones, to represent differences in rock-mass structure.

**Table 5. Schmidt hammer sampling results and the estimated geomechanical properties that supports the rock-mass quality analysis. ID = Identification; Dip Dir = Dip direction; SD = Standard Deviation; JCS = Joint Compression Strength; ϕr = Residual friction**

**angle; ϕp = Peak friction angle.**

| ID | Structural Geology | | Schmidt hammer sampling | | | | Geomechanical properties | | |
|---|---|---|---|---|---|---|---|---|---|
| | Dip Dir | Dip | Average Rebound (Q) value | Highest | Lowest | SD | JCS (MPa) | ϕr (°) | ϕp (°) |
| 1 | 130 | 90 | 66.4 | 70 | 57.5 | 11 | 164.2 | 25.6 | 61.4 |
| 2 | 160 | 90 | 70.1 | 86 | 67 | 10.8 | 217.3 | 26.5 | 63.2 |
| 3 | 10 | 30 | 66.7 | 84 | 50 | 13.4 | 167.9 | 25.7 | 61.5 |
| 4 | 350 | 25 | 50.4 | 55 | 40 | 7.9 | 61.3 | 21.8 | 54.2 |
| 5 | 262 | 80 | 64.4 | 67.5 | 61.1 | 9.1 | 140.9 | 25.1 | 60.4 |
| 6 | 330 | 85 | 75.3 | 80 | 70 | 11.5 | 250 | 27.7 | 64.9 |
| 7 | 185 | 20 | 51.4 | 68 | 37 | 11.1 | 64.8 | 22.1 | 54.6 |

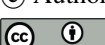



| 8 | 50 | 75 | 72.4 | 78 | 67 | 10.6 | 194.8 | 27 | 63.4 |
|---|---|---|---|---|---|---|---|---|---|
| 9 | 60 | 5 | 66.2 | 81 | 52 | 11.5 | 161.3 | 25.6 | 61.3 |
| 10 | 190 | 90 | 28.3 | 42 | 11 | 9.6 | 20.4 | 16.7 | 45.2 |
| 11 | 25 | 15 | 19.9 | 29 | 0 | 5.8 | 12.3 | 14.7 | 41.4 |
| 12 | 10 | 15 | 76.1 | 80 | 50 | 13.4 | 241.2 | 27.9 | 65 |
| 13 | 15 | 20 | 69.3 | 78 | 50 | 12.2 | 211.4 | 26.3 | 62.9 |
| 14 | 25 | 15 | 3.1 | 13 | 0 | 5.8 | 1.5 | 10.7 | 30.2 |
| 15 | 346 | 15 | 34.9 | 44 | 0 | 9.4 | 28.6 | 18.2 | 47.9 |
| 16 | 352 | 5 | 4.3 | 13 | 0 | 4.9 | 2.2 | 11 | 31.7 |
| 17 | 350 | 5 | 31.7 | 45 | 27 | 6.2 | 24.3 | 17.5 | 46.6 |
| 18 | 328 | 85 | 68.7 | 77 | 58 | 9.8 | 199.7 | 26.2 | 62.6 |
| 19 | 260 | 75 | 27.3 | 61 | 19 | 8.9 | 19.2 | 16.4 | 44.7 |
| 20 | 260 | 75 | 67.6 | 73 | 57 | 9.5 | 180.8 | 25.9 | 62 |
| 21 | 260 | 75 | 83.3 | 88 | 75 | 9.8 | 260.4 | 29.6 | 66.9 |
| 22 | 28 | 8 | 60.4 | 65 | 54 | 7.2 | 107.9 | 24.2 | 58.5 |
| 23 | 193 | 75 | 71.9 | 82.5 | 54 | 11 | 233.1 | 26.9 | 63.9 |
| 24 | 193 | 75 | 62 | 80.5 | 34.5 | 16 | 119.6 | 24.6 | 59.2 |
| 25 | 350 | 5 | 12 | 17.5 | 0 | 7.3 | 6.6 | 12.8 | 37.4 |
| 26 | 5 | 10 | 33.5 | 39.5 | 15 | 7.2 | 26.6 | 17.9 | 47.3 |
| 27 | 53 | 85 | 25.1 | 47 | 9.5 | 8 | 17 | 15.9 | 43.8 |
| 28 | 152 | 85 | 35.4 | 48 | 16 | 4.6 | 29.2 | 18.3 | 48.1 |
| 29 | 320 | 75 | 15.4 | 28 | 12.5 | 4.9 | 8.9 | 13.6 | 39.2 |
| 30 | 160 | 35 | 45.7 | 55 | 33 | 7.6 | 48.5 | 20.7 | 52.3 |
| 31 | 240 | 85 | 64.1 | 71 | 45 | 8.5 | 138.5 | 25.1 | 60.2 |
| 32 | 220 | 50 | 55.5 | 69 | 44 | 8.7 | 80.8 | 23.1 | 56.3 |
| 33 | 330 | 75 | 15.9 | 30 | 0 | 8.1 | 9.2 | 13.7 | 39.5 |

**Table 6. Estimation of the Uniaxial Compression Strength (UCS), based on the Schmidt Hammer sampling campaign, and corresponding weathering degree of the bedrock in the canyon area.**

| Weathering degree | Schmidt Hammer Rebound (Q) | | Uniaxial Compression Strength - UCS (MPa) | | Residual friction angle | Peak friction angle |
|---|---|---|---|---|---|---|
| | Average Rebound | Interval | Estimates | Interval | Estimates | Estimates |
| I | 76.6 | > 71 | 246.2 | > 230 | 28 | 65.2 |
| II | 68.4 | 71 - 66 | 187.2 | 230 - 160 | 26.1 | 62.3 |
| III | 58.3 | 66 - 50 | 102 | 160 - 60 | 23.7 | 57.6 |
| IV | 32.7 | 50 - 25 | 26.7 | 60 - 20 | 17.7 | 47 |
| V | 11.8 | < 25 | 6.8 | < 20 | 12.8 | 36.6 |





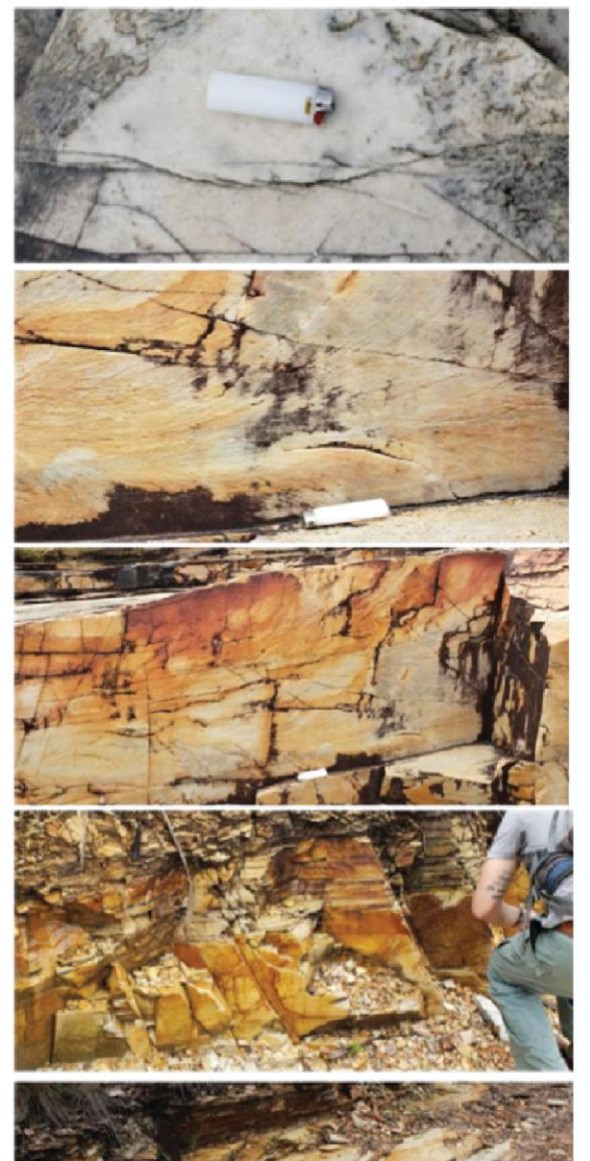

## Weathering degree I - Fresh rock

No weathering evidence and strong resistance to geological hammer.

## Weathering degree II - Slightly weathered

Weathering evidences in rock matrix (e.g., opaque, slightly discolored minerals, impregnated with Fe oxides-hydroxides). Brittle after several impacts with geological hammer.

## Weathering degree III - moderately weathered

Weathering evidences in rock matrix (e.g., discol-ored, opaque minerals, impregnated with Fe oxides-hydroxides). Few impacts with geological hammer brittle the rock mass.

## Weathering degree IV - very weathered

Very weathered minerals in rock matrix, opaque and discolored, impregnated with Fe oxides-hydroxides. Geological hammer easily brittles the rock mass.

## Weathering degree V - Saprolite

Very altherated minerals that brittle very easily. Rock-mass structure is easily desintegrated with the pres-sure of hands. It is only observed in more micaceous facies of the bedrock.

**Figure 10: Different weathering degrees observed in the bedrock.**

The geomechanical compartment C1 corresponds to the wall where the rock-toppling event occurred, with N41W and N25E strikes. This compartment was subdivided into four (4) structural zones (**Figure 11B**):





- • C1.1: rated as a "Very Weak" rock mass (Final Rating: 14), as it shows a high density of joints and fractures, as well as a high weathering degree (saprolite). Water infiltrates and percolates through the joints, where soil and vegetation are formed. This structural zone is located in the intermediate portion of the rock wall, characterized by a slightly more micaceous facies in the Furnas Quartzite.

- • C1.2: rated as "Very Weak" rock mass (Final Rating: 20), corresponding to the portion where the rock-toppling event occurred. The spacing between joints is narrow (< 30 cm) (**Figure 9C**), with water percolating in rainy periods. The joints can show a large aperture (up to 50 cm) in this structural zone, which may be filled with soil and vegetation that can favor the occurrence of slope failure. The fault zone with NW/SE and NE/SW direction heavily influences the geomechanical behavior of this structural zone;

- • C1.3: rated as a "Very Weak" rock mass (Final Rating: 20), due to continuous and persistent joints in the bedrock, with the occurrence of soil and vegetation in joint planes that can potentially accelerate slope failures. This compartment is located in the upper portions of the canyon wall;

- • C1.4: rated as "Normal" rock mass (Final Rating: 52), as the joints in this portion are more spaced from one another (> 60 cm), also showing a much narrower aperture, making this portion less prone to instabilities.

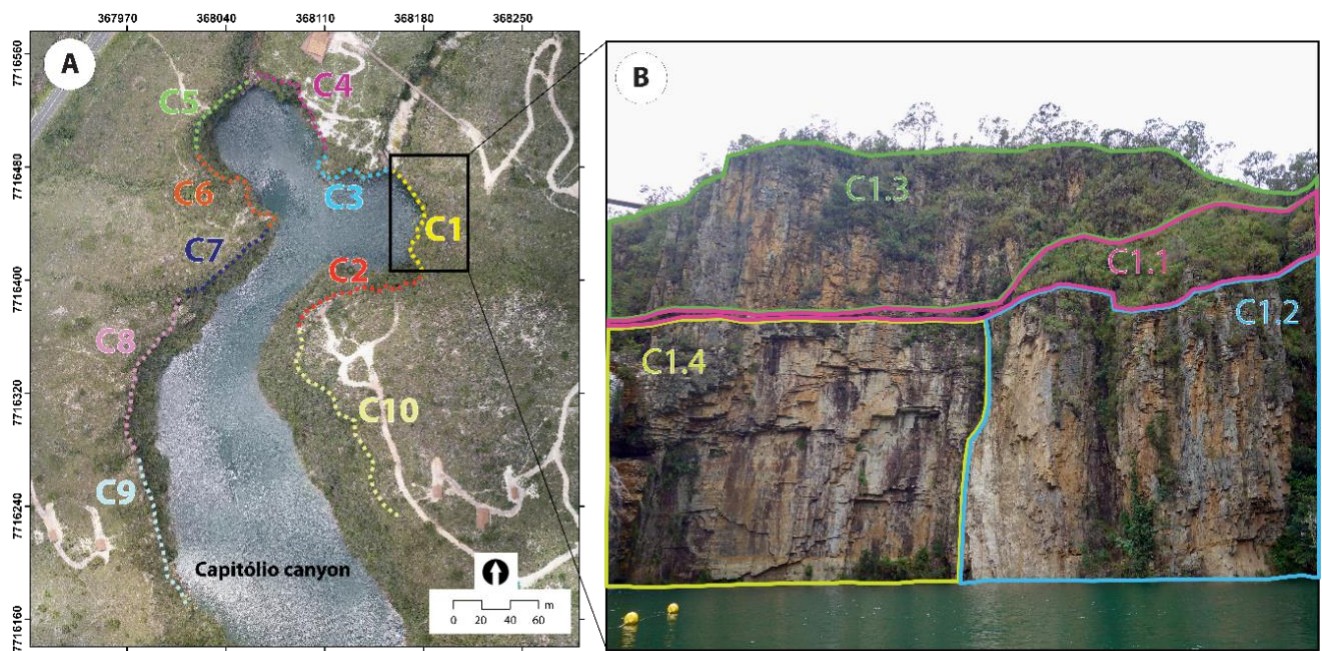

**Figure 11. Geomechanical classification of the rock mass in the Capitólio canyon. A) Ten compartments were individualized in the canyon, representing each canyon wall. B) In each canyon wall, the rock mass was further compartmentalized in smaller structural zones, due to differences in the bedrock properties that translate into different geomechanical behaviour.**

The geomechanical characteristics of each compartmented zone is shown in **Table 7**, as well as the final ratings of the rock mass. The classification process was repeated for all the walls in all of the four canyons and, basically, all the walls in the analyzed canyons exhibited similar characteristics to those described in these four structural zones of compartment C1.



The classification of the rock mass quality in the canyons, with the subdivision in smaller structural zones, was essential to
identifying potential areas in the bedrock where rockfall and rock-toppling processes are more likely to occur in the future,
especially considering that no historical database is available for the study site.

Areas that show high susceptibility to rockfall and rock slides are basically the upper portions of the canyons, where structural
zone C1.3 occurs, with smaller mobilizable blocks of up to $1 - 2$ m$^3$. Portions in the canyons where higher-magnitude rockfall
and rock-toppling can potentially occur, such as the event of 2022, are mostly related to intersection of the NW-SE and NE-
SW fault zones (**Figure 9B**) which are also areas that the bedrock shows a generally poorer geomechanical quality, with
characteristics similar to that of the structural zone C1.2 (**Table 7**).

**Table 7: Rock-mass rating of each structural zone in the canyon wall where the fatal rock-toppling event occurred.**

| Geomechanical parameters | Structural zone | | | |
|---|---|---|---|---|
| | **C1.1** | **C1.2** | **C1.3** | **C1.4** |
| Uniaxial Compression Strength (UCS) | 1 - 15 MPa | 15 - 25 MPa | 15 - 25 MPa | 100 - 250 MPa |
| **Score** | **1** | **2** | **2** | **12** |
| Discontinuities/joints per meter | 13 | 14 | 15 | 4 |
| **Score** | **20** | **19** | **18** | **22** |
| Discontinuities/joints condition | Continuity > 10 m, slickensided, soft gouge infilling > 5 mm, decomposed | Continuity > 10 m, smooth, soft gouge infilling > 5 mm, highly weathered | Continuity > 10 m, smooth, soft gouge infilling < 5 mm, highly weathered | Continuity > 10 m, smooth, Hard gouge infilling < 5 mm, moderately weathered |
| **Score** | **0** | **1** | **3** | **8** |
| Water flow | Dripping | Humid | Dripping | Slightly Humid |
| **Score** | **4** | **4** | **7** | **10** |
| Alterability | <30 | 30 - 60 | <30 | > 85 |
| **Score** | **0** | **4** | **0** | **10** |
| $F_o$ | Unfavorable | Unfavorable | Unfavorable | Unfavorable |
| F1 | *1* | *1* | *1* | *1* |
| F2 | *1* | *1* | *1* | *1* |
| F3 | *-25* | *-25* | *-25* | *-25* |
| F4 | *15* | *15* | *15* | *15* |
| $F_0$ **score** $[F_0 = (F_1 * F_2 * F_3) + F_4]$ | **-10** | **-10** | **-10** | **-10** |
| **RMR14:** | **14** | **20** | **20** | **52** |
| **Rock-mass Rating** | **Very Weak** | **Very Weak** | **Very Weak** | **Normal** |





The kinematic analysis further supports these observations and interpretations, providing an overview of the susceptibility to different types of slope failure considering each geomechanical compartment. Overall, the canyons show a high susceptibility

to planar failures (**Figure 12A**), and low susceptibility to both wedge failure and toppling (**Figure 12B** and **12C**, respectively). However, when certain slope face directions are considered, some portions of the canyons can show a higher susceptibility to toppling (**Table 8** and **9**).

Geomechanical compartments with slopes faces with a dip direction/dip of 142/90 (compartment C5), 135/90 (C7), 125/90 (C8) and, secondarily, 54/90 (C6) tend to show higher susceptibility to toppling when compared to other slope faces (**Table**

**9**). These compartments are notable due to the formation of rock columns (**Figure 13A**), up to 50 m high, which are areas on the canyons that already show movement signs in the slopes. Moreover, considering that the slope faces in the canyons are characterized by abrupt changes in strikes, more localized unfavorable conditions can occur in other compartments as well.

**Table 8: Kinematic analysis results for each geomechanical compartment in the Capitólio canyon, considering rock-toppling susceptibility. $\phi r$ = Residual friction angle; $\phi p$ = Peak friction angle.**

| Rock-toppling kinematic analysis | | | | | | | | | |
|---|---|---|---|---|---|---|---|---|---|
| **Parameters** | | **Geomechanical compartments** | | | | | | | |
| | | **C1** | | **C2** | | **C3** | | **C4** | |
| **Slope direction** | | 229/90 | 295/90 | 330/90 | 350/90 | 180/90 | 225/90 | 315/90 | 240/90 | 195/90 |
| **Unweathered bedrock** | $\phi p$= 65.2 | 6.4% | 7.9% | 7.4% | 11.4% | 13% | 6.9% | 7.4% | 3.9% | 8.3% |
| | $\phi r$ = 28.0 | 6.7% | 10.5% | 9.8% | 12.3% | 13.5% | 7.2% | 10.2% | 3.9% | 8.8% |
| **Weathered bedrock** | $\phi p$ = 36.6 | 6.5% | 10.5% | 9.7% | 12.3% | 13.3% | 7.11% | 10.2% | 3.9% | 8.8% |
| | $\phi r$ = 12.8 | 6.7% | 10.5% | 9.8% | 12.3% | 14% | 7.2% | 10.2% | 3.9% | 9.3% |


**Table 9: Kinematic analysis results for each geomechanical compartment in the Capitólio canyon, considering rock-toppling susceptibility. $\phi r$ = Residual friction angle; $\phi p$ = Peak friction angle.**

| Rock-toppling kinematic analysis | | | | | | | | | |
|---|---|---|---|---|---|---|---|---|---|
| **Parameters** | | **Geomechanical compartments** | | | | | | | |
| | | **C5** | | **C6** | **C7** | | **C8** | **C9** | **C10** |
| **Slope direction** | | 142/90 | 92/90 | 54/90 | 135/90 | 95/90 | 125/90 | 80/90 | 230/90 |
| **Unweathered bedrock** | $\phi p$= 65.2 | 19.2% | 11.6% | 24.4% | 20.4% | 11.7% | 19.3% | 13.3% | 6% |
| | $\phi r$ = 28.0 | 20.1% | 13.3% | 27% | 21.1% | 13.5% | 20.6% | 15.4% | 6.2% |
| **Weathered bedrock** | $\phi p$ = 36.6 | 20.10% | 13.3% | 27% | 21.1% | 13.5% | 20.6% | 15.4% | 6% |
| | $\phi r$ = 12.8 | 20.4% | 13.3% | 27.2% | 21.1% | 13.5% | 20.6% | 15.6% | 6.2% |





**Figure 12: Kinematic analysis results using Rocscience DIPS. A) Planar failure. B) Wedge failure. C) Toppling failure.**



Abrupt changes in the direction of slope faces can be observed, for instance, in compartment C1, where the 2022 event occurred, with the intersection of NW-SE (dip dir/dip = 229/90) and NE-SW (dip dir/dip = 295/90) slope faces (**Figure 8C**). These areas, as highlighted earlier, are also associated with the intersection of the NW-SE and NE-SW fault zones. Compartment C2 also show a similar condition (**Figure 13A**), with a rock column of 20 m showing signs of movements, which can potentially experience a similar fate to the slope that toppled on January 2022.

Finally, based on the information about rock-mass quality and our field investigations, areas with high susceptibility to landslide occurrence (**Figure 13**), where constant monitoring should be conducted, were identified and delimitated in all four canyons: Capitólio (**Figure 13A**), Tucanos (**Figure 13B**), Cascatinha (**Figure 13C**) and Cabritos (**Figure 13D**). These areas are characterized by a bedrock with poor mechanical quality (C1.1, C1.2 and C1.3) and unfavorable structural geology, considering the relationship of slope face and bedrock discontinuities.

The implementation of retention structures and other safety measures, such as destruction of the rock body and/or manual acceleration of the slope movement (i.e., hazard elimination practices) can potentially be implemented by public authorities and park administrators, although a more long-term analysis is needed to assess their efficacy. The isolation of areas that already show slope movements signs and keeping a safety distance from the canyons, especially where the rock-mass quality is weaker, can also support a safer visitation experience, without impacting the landscape. The location of the sites that should

be monitored in the four canyons is available as Supplementary Information (Figure S1).





**Figure 13: Overview of areas with high landslide susceptibility in the four canyons. A) In the Capitólio canyon, very fractured rock columns are observed, with aperture of up to 50 – 60 cm. This high susceptibility scenario supports the restriction of the core area of the canyon to visitation. B) In the Tucanos canyon, high susceptibility scenarios near the entrance of the canyon are observed. C) In the Cascatinha canyon, several high susceptibility scenarios are observed along the access to the waterfall located at the end of the canyon. In this example, the rockfall hazard is high. D) The Cabritos canyons is located near the Furnas dam. The canyon wall can reach up to 100 m high and very large rock columns show instability signals, such as the inclination towards the canyons area (left ellipse) and fractures with very large aperture - up to 1 m (right yellow ellipse).**






## 5 Discussions

Compared to the results of the kinematic analysis, the rock-mass quality evaluation provided a more specific delimitation of the areas that are prone to rock-toppling and landslides (i.e., rockfall and rockslides), which, in the context of Geotourism, is more useful, as it points out specific locations where risk management strategies are required, supporting a more objective evaluation if a structural (e.g., retention structures) or a non-structural (e.g., access restriction) solution is more adequate.

As landslide events in rock slopes are mainly conditioned by their geomechanical properties and structural geology, the 440 application of methodologies that analyze rock-mass quality can successfully support their integration. Other studies have also applied different rock-mass quality methodologies in landslide hazard assessments, from shallow and rotational landslides (e.g., Brideau et al., 2007; Lee et al., 2018; Siddique et al., 2020; Das et al., 2024) to rockfall and rock-toppling (e.g., Matasci et al., 2018; Wollenberg-Barron et al., 2023; Cackrabuana et al., 2024), highlighting their wide applicability. A limitation of these methodologies is that they can be very subjective, as well as dependent on detailed field observations (e.g., structural 445 geology and visual analysis), which can be expensive and time consuming.

The adapted RMR14 methodology presented in this study is not radically different from the SMR developed by Romana (1993), although it incorporates some of the improvements presented in RMR14 in relation to the RMR89, which is a better correlation to other methodologies that analyze rock-mass geomechanical properties (Zhang et al., 2019). Moreover, as highlighted by studies such as Cackrabuana et al. (2024) and Zhang et al. (2019), the application of RMR14 is relevant to 450 demonstrate that the update is effective in representing a rock-mass' geomechanical behavior in different contexts, not only on tunneling or mining projects.

Our study is resulted from several months of field investigations and data analysis, as well as years (approx. 2 years) of discussions with local stakeholders and workers about the feasibility of a particular hazard assessment methodology that can support risk-reduction strategies. The application of the adapted RMR14 is an easy to comprehend and replicate method that 455 help to determine rockfall and rock-toppling susceptibility, which can be applied in other tourist areas of the reservoir, such as waterfalls and rock slopes used for climbing.

While the rock-toppling event in January 2022 was resulted from a combination of factors (e.g., erosion, water level fluctuations, rainfall), the adapted RMR14 method was fundamental in supporting the delimitation of areas that should be avoided and areas where tourists should not be exposed to for long periods of time. It is important that Geodiversity can be 460 seen an asset that can bring economic development, especially when it can be performed in a safely manner, so the methodological steps applied in our study considered both the preservation of the Geological heritage and the maintenance of their use for tourism.

Finally, a recent study conducted remotely in the region by Sun et al. (2024) performed a numerical simulation of the rock-toppling event, based on high-resolution data (DEM) provided by the hydropower plant company. Sun et al. (2024) also suggest 465 that the rock-toppling event was caused by rain infiltration and weakened support at the base of the slope, although water-level fluctuations is not considered in their analysis nor a detailed structural geology analysis is performed, which, as our study



suggests, are important factors that contribute to slope failures in the region. Moreover, they attribute that the erosion at the base of canyon wall is related to the occurrence of sedimentary carbonate rocks underneath the Furnas Quartzite, which, considering that we are within a complex metamorphic setting (Passos *Nappe*), is very unlikely and not observed by us during 470 our extensive field investigations.

## 6 Conclusions

The 2022 rock-toppling event in the Furnas reservoir can be attributed to a combination of different factors, mainly the erosion of the bottom part of the slope, rainfall, and fluctuations in reservoir water level. The very fractured bedrock, characterized by three families of joint systems and two perpendicular fault zones, is also an important predisposing factor. Rainfall infiltration 475 in the bedrock's discontinuities, combined with a rapid increase in reservoir water level, is interpreted to have contributed to the event's initiation, even though precipitation records were not particularly intense.

Differences in the rock-mass geomechanical behavior significantly contribute to slope failures; therefore, the use of a geomechanical classification method, such as the RMR14 adapted for rock slopes, was fundamental to identifying the areas in the bedrock that are most prone to landslides. The orientation of the canyon walls in relation to the bedrock discontinuities is 480 an important parameter of rockfall and rock-toppling susceptibility studies and the incorporation of this parameter in the RMR14 methodology adapted for open rock slopes studies, following what is proposed in the SMR, can successfully be replicated in future studies.

Considering areas that are dependent on Geotourism and are susceptible to rockfall and rock-toppling, the inclusion of rock-mass quality analysis is important in the delimitation of areas that should be restricted to visitation or where retention structures 485 are needed. Their inclusion in a systematic hazard analysis methodology is valuable, as it is more successful in precisely delimitating landslide-prone areas compared to kinematic analysis, as it integrates bedrock's geomechanical behavior and its structural geology.

Furthermore, this fatal rock-toppling event highlighted the need for geohazard-management strategies focusing on tourist areas. The suggestion of retention structures or hazard elimination practices (i.e., demolition of hazardous rock blocks) in areas with 490 higher susceptibility to landslides is meaningful to protecting visitors and workers, within the scope of a geological risk management system that includes a geological-geotechnical monitoring program. However, as geoscientists, we should also consider the option of not interfering with the natural environment, as natural-disaster risk is an intrinsic factor in various types of tourism.



## Declarations

### Author contribution

**Victor Cabral**: writing, data collection, data analysis, funding. **Fabio Augusto Gomes Vieira Reis**: field campaigns, funding, data analysis. **Joana Paula Sanchez**: Data analysis, field campaigns, writing. **Rodrigo Irineu Cerri**: data collection, data analysis, writing. **João Paulo Monticelli**: data collection, data analysis, writing. **Claudia Vanessa dos Santos Correa and Vinicius Queiroz Veloso**: data analysis, writing. **Débora Moraes Duarte, Guidotti de Souza dos Garion, George A. Longhitano, Bruno Fructuoso Coelho de Souza, Marcelo Fischer Gramani and Caiubi Emanuel Kuhn**: Data collection. **Lucilia do Carmo Giordano**: data analysis, writing.

### Competing Interests

The authors declare that they do not have any competing interests.

### Financial Support

This study was financed in part by the São Paulo Research Foundation (FAPESP, 2023/02458-6), Brazil's National Council for Scientific and Technological Development (CNPq, 316574/2021-0) and the Federal Agency for Support and Evaluation of Graduate Education (CAPES, 88881.705011/2022-01).

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
