# Peer review of "Geological-geotechnical analysis of a rock-toppling prone canyon in Furnas, Brazil, after a fatal event"

_EGUsphere, 2025_

## Author Comment (AC1)

**Point-by-point response to comments made by Reviewer #2**

1. *Readers would like to know if the kinematic criterion for toppling was met because toppling can be caused by the deformation of a weak base of a column, which progressively meets the conditions for toppling. A precise cross-section of the geology of the fallen column must be provided, including the locations of the discontinuities.*

**Unfortunately, elevation data from the canyons before the event was not made available to our team despite our requests to the company that operates the hydropower plant. This request was also made by the city of Capitólio, which has also never received the data.**

**In this context, we performed a stability analysis and a more detailed kinematic evaluation in a portion of the Capitólio canyon that is interpreted to have a similar context to the location where the rock-toppling event occurred. While this portion did not fail on January 8, 2022, the variations in the FoS can support our analysis of the conditions that potentially led to the fatal event.**

**The stability analysis is based on the combination of FEM and LEM methods, to support the investigation of the effect that some previously inferred mechanisms had on rock instability, such as rainfall infiltration and reservoir water-level fluctuations. The FEM analysis evaluated the effect of reservoir water-level fluctuations and rainfall infiltration on the rock mass, with the results (pore pressure) being imported in our LEM analysis of the slope stability.**

**The updated kinematic analysis was conducted in Stereonet.**

**The methodology of the stability analysis is described as follows:**

"Once the probabilities of the failure(s) mode(s) have been determined, the next step was the stability analysis considering the 2022 event (toppling), which compares the ratio of forces resisting failure with shear forces (i.e., the Factor of Safety – FoS) (Norrish and Wyllie, 1996). As highlighted previously, elevation data from the canyons before the event was not made available to our team despite our requests to the company that operates the hydropower plant, so the stability analysis is conducted in a portion of the Capitólio canyon (Figure 5) that is interpreted to have a similar geological-geotechnical context to the location where the rock-toppling event occurred. While this portion did not fail on January 8, 2022, the variations in FoS can support our analysis of the conditions that led to the fatal event.

For the implementation of the FEM-LEM-combined method in the stability analysis, Seep/W (FEM) and Slope/W (LEM) are applied. Slope/W imports pore-water pressure data from Seep/W, supporting the analysis of seepage on rock instabilities in the canyon. The Method of Morgernstern-Price (M-P) is used in Slope/W, as it places no restriction on the shape of the failure surface, thus more adequate to rock slopes, satisfying both force and moment equilibrium.

As proposes Yin et al. (2016), first we consider the steady-state condition as the initial condition for the next analysis, which is when rainfall and reservoir water level fluctuations are included, both separately and combined. The Factor of Safety (FoS) in Slope/W is calculated based on the transient seepage condition resulted from Seep/W. The physical-mechanical parameters applied in the calculation are based on in situ measurements and estimations using the Schmidt Hammer, as well on literature data from Vidal et al. (2014), Lògó and Vásárhelyi et al. (2019) and Sujatono and Wijaya (2022) for Quartzites. For simplicity, strength parameters of the rock slope in natural and wet state are assumed to be the same (Tang et al., 2015).

The hydraulic conductivity (K) for a joint set of the rock mass is estimated based on the equation from Snow (1968) for fractured systems with parallel array of planar fractures (Eq. 7). The aperture of the fractures (b), and the number of fractures per unit distance of rock face (N) are required to estimate K, which are parameters estimated in our field campaigns. Other parameters are water density ($\rho$), the gravitational constant (g) and the dynamic viscosity ($\mu$).

$$K_{joint\ set} = \frac{\frac{\rho g}{\mu} N b^3}{12} \qquad (7)$$

For the seepage analysis, we assume that if the elevation point is above reservoir level, the flux is zero and other far-field boundaries are assumed as non-permeable (Yin et al., 2016). The rainfall-induced seepage was simulated by applying a unit flux as no-flow boundaries.

On Seep/W, we first establish the rock-toppling area and define the rock properties, to proceed with the definition of the groundwater/reservoir head boundary. After these first two steps, the entry and exit range of the failure surface is set on Slope/W, to calculate the FoS. The slope stability is analyzed in the following three situations:

- **Under a fluctuating water level condition**: As a reservoir water level fluctuates, it can impact the pore water pressure in the discontinuities, as well as potentially weaken geomechanical properties of the bedrock, facilitating erosion (Tannant et al., 2017).

- **Under rainfall conditions**: As the rainfall continues and intensifies, the infiltration of rainwater into the fractured rock mass can influence hydrostatic and hydrodynamic pressure of the slope, with the accumulation of water in the discontinuities increasing bulk density and decreasing shear resistance. Water infiltration will also impact groundwater level, reducing effective normal stress (Tannant et al., 2017).

- **Water-level fluctuations + rainfall**: Rainfall can increase bulk density and decrease shear resistance, while a fluctuating water level can impact the pore water pressure in the discontinuities, as well as the resistance to erosion of the rock mass.

[Figure]

Figure 5: Stability analysis. A) Overview of the Capitólio canyon showing the location where the stability analysis is conducted (B) and the site where the rock-toppling event occurred (C). This picture is from before the event (February, 2019), when reservoir water level was much lower (around 755 m asl) than in February and March, 2022, when our field campaigns were conducted (765 m asl). (Source: Ion David Zarantonelli, 2019). B) Location where the analysis is conducted, interpreted as having a similar geological-geotechnical condition as the location of the rock instability. B) Post-event profile photo of the location where the rock toppling occurred.

**New references:**

Lógó, B. A., Vásárhelyi, B. Estimation of the Poisson's Rate of the Intact Rock in the Function of the Rigidity. Periodica Polytechnica Civil Engineering, 63(4), pp. 1030–1037, 2019. https://doi.org/10.3311/PPci.14946

Morgenstern N R and Price V E 1965 The analysis of the stability of general slip surfaces; Geotechnique 15(1) 79–93.

Sujatono, S., Wijaya, A.E. The influence of quartz content on modulus of elasticity and Poisson's ratio in quartz sandstone. Bull Eng Geol Environ 81, 287 (2022). https://doi.org/10.1007/s10064-022-02798-6

Tang, H., Li, C., Hu, X. et al. Deformation response of the Huangtupo landslide to rainfall and the changing levels of the Three Gorges Reservoir. Bull Eng Geol Environ 74, 933–942 (2015). https://doi.org/10.1007/s10064-014-0671-z

Tannant, D. D., Giordan, D., & Morgenroth, J. (2017). Characterization and analysis of a translational rockslide on a stepped-planar slip surface. Engineering Geology, 220, 144-151. https://doi.org/10.1016/j.enggeo.2017.02.004

Vidal, FH.; Castro, NF; Azevedo, HCA. Tecnologia de rochas ornamentais: pesquisa, lavra e beneficiamento – Rio de Janeiro: CETEM/MCTI, 700p.: il. 2013.

Yin, Y., Zhang, L., Liu, Y., & Wang, H. (2016). Reservoir-induced landslides and risk control in Three Gorges project on Yangtze River, China. Journal of Rock Mechanics and Geotechnical Engineering, 8(5), 577–595. https://doi.org/10.1016/j.jrmge.2016.08.001

**From the results of the stability analysis, considering the eight-day period prior to the disaster (January 1 – 8, 2022), we could observe that the Factor of Safety (FoS) of the slope was more impacted by the combination both rainfall and reservoir water-level increase, with rainfall suggested as having a slightly more influence over the overall stability than the water-level fluctuations, when they were analyzed separately.**

**These results are detailed in the revised version of manuscript, in the new subchapter "4.3. Stability analysis and the factors that influence slope stability in the canyons". This subchapter incorporated the results of the previous version of the manuscript as well, corroborating the inferred data that was originally presented.**

"Thus, to quantitatively analyze the influence of the reservoir water-level fluctuations and rainfall on the slope stability, we employed Seep/W considering the period of eight days prior to the rock-toppling event, from January 1 to January 8. First, the dynamic seepage field is calculated based on FEM with the varying water level and rainfall data. The partitions of the material are then reassigned at every step with the dynamic seepage field, so that the FoS can be finally computed by the Morgernstein-Price Method on Slope/W (LEM) (**Figure 13**). The physical-mechanical parameters (**Table 12**) applied in the stability analysis are based on field surveys and geomechanical investigation of the rock mass (shear resistance parameters and hydraulic conductivity), while the Poisson's ratio and Elastic Modulus are based on average data for Quartzites (Sujatono and Wijaya, 2022; Lógó and Vásárhelyi, 2019).

The stability analysis is conducted in a site in the Capitólio canyon that is similar to the one where the rock-toppling occurred (**Figure 5A** and **Figure 13A**), as no high-resolution data from before the 2022 event was made available to our team. As in the date of our photogrammetric data acquisition (May 2022) the reservoir water level was different from the day of the event (January 2022), we performed our seepage analysis using FEM assuming a daily increase similar to what occurred in the days leading to the disaster (**Table 11** and **Figure 13B**).

Table 11: Evolution of the Factor of Safety (FoS) considering the effect of rainfall, reservoir water level fluctuations and the combination of both. The rock-toppling event occurred on January 8, 2022. The stability analysis considers a site in the canyon that is interpreted to have a similar geological-geotechnical condition as the one where the slope failure occurred, since no pre-event data was made available to our team.

| Rainfall (mm) | Reservoir water level | Reservoir water level | Factor of Safety analysis |
|---|---|---|---|

| Date (dd/mm/yyyy) | | on the days leading to the rock-toppling event (m asl) | assumed in the stability analysis (m asl) | FoS Reservoir level | FoS Rainfall | FoS Rain+reservoir |
|---|---|---|---|---|---|---|
| 01/01/2022 | 21 | 757.48 | 766.16 | 1.309 | 1.309 | 1.309 |
| 02/01/2022 | 24 | 757.64 | 766.32 | 1.305 | 1.306 | 1.305 |
| 03/01/2022 | 33 | 757.81 | 766.49 | 1.302 | 1.302 | 1.302 |
| 04/01/2022 | 5 | 758.01 | 766.69 | 1.302 | 1.3 | 1.299 |
| 05/01/2022 | 18 | 758.21 | 766.89 | 1.301 | 1.298 | 1.297 |
| 06/01/2022 | 2 | 758.39 | 767.07 | 1.301 | 1.298 | 1.297 |
| 07/01/2022 | 27 | 758.64 | 767.32 | 1.299 | 1.297 | 1.294 |
| 08/01/2022 | 8 | 758.87 | 767.55 | 1.299 | 1.297 | 1.293 |

Table 12: Physical-mechanical parameters applied in the rock slope stability analysis.

| Material | Unit Weight (kN/m³) | Elastic modulus (Gpa) | Poisson's ratio | Shear Strength | | Hydraulic conductivity (m/s) |
|---|---|---|---|---|---|---|
| | | | | Cohesion (kPa) | Internal Friction Angle (°) | |
| Bedrock (Furnas Quartzite) | 25 | 20 | 0.18 | 2000 | 28 | $6 \times 10^{-4}$ |

[Figure]

Figure 13: Slope stability analysis. A) Cross-section of the location interpreted as having a similar geological-geotechnical characteristic as the site that failed on January 8, 2022. The Stereonet highlights that this portion is prone to toppling, especially considering the main tension crack in the slope. B) 3D stability analysis on Slope/W from Geostudio (Seequent, 2025), considering reservoir waterl-level fluctuations and the structural geology.

When only water-level fluctuations were considered, the numerical results show that it impacted slightly the FoS of the slope, decreasing from 1.309 to 1.299 in the eight-day period (**Figure 14 and Table 11**). The impact of rainfall on the FoS was slightly higher, decreasing to 1.297 from 1.309 in the considered timeline. The combined effect of rainfall and reservoir water-level fluctuation impacted the FoS the most, decreasing from 1.309 to 1.293. While these scenarios were not sufficient to cause a slope failure at this location, it can suggest how these different factors impacted the site that did fail on January 2022. Thus, based on this relationship between FoS, water-level variations and precipitation, it can be suggested that both rainfall and reservoir water level had a role on the rock-toppling initiation, with rainfall infiltration with a slightly stronger influence on slope stability.

Furthermore, during our field surveys, it was observed the formation of cavities at the bottom of the slope where rock-toppling event occurred, caused by the long-term erosion in this region (**Figure 15A and 15B**). These cavities are interpreted to be one of the main predisposing factors to slope failure and are potentially resulted from the effect of waves in the reservoir, caused by wind and boat circulation, as well as by water flow from the nearby waterfall (**Figure 15A**). Moreover, the long-term fluctuations in the reservoir water level (**Figure 11A**) are also inferred as one of the contributors to the formation of these cavities, weakening the geomechanical properties of the bedrock (Tannant et al., 2017). These cavities caused the middle section to lose underlying support, with an added pressure of water in the joints. As pointed by Cruden (1991), self-weight alone is not sufficient to cause toppling, with external forces needed to initiate the process. In the site that we performed the stability analysis, these cavities were not observed (**Figure 15C**), which can contribute to increasing the stability of that portion of the canyon.

Figure 14: Evolution of Factor of Safety based on the effect of rainfall, reservoir level increase and the combination of both processes.**”**

**Moreover, the results of a more detailed kinematic analysis of the site that failed on January 8, 2022, and of the site that we performed our stability analysis is presented on the revised subchapter about the kinematic Analysis. Also, Figure 13 (presented above) demonstrates that the kinematic criterion for toppling is met in the stability-analysis site, considering the main tension crack that is observed in the slope.**

**"[…]**

Although not showing an overall high susceptibility to toppling in the kinematic analysis, probably due to the generalization of the slope-face direction, Compartment C2 shows a very high joint frequency (**Figure 5B** and **Figure 7A**) and signs of slope movements, with a tilting 20 m rock column that can potentially experience a similar fate to the slope that toppled on January 2022. The kinematic analysis of this section in Compartment C2, which is interpreted to have a similar condition to the site where the 2022 rock-toppling event occurred, further highlights that this specific region has a high susceptibility to toppling, with a similar probability to this geodynamic process as the slope-failure site (**Table 10** and **Figure 10**).

[Figure]

Figure 10: Kinematic analysis of the rock-toppling site (left) and the site that is interpreted as having a similar geological-geotechnical condition as the one that failed on January 2022 (right). The kinematic analysis was conducted considering weathered and unweathered portions of the rock mass, as well as both the peak and residual friction angle.

Table 10: Kinematic analysis results for the rock toppling site and the site that is interpreted as having a similar geological-geotechnical condition as the site that failed on January 2022. $\phi r$ = Residual friction angle; $\phi p$ = Peak friction angle. The percentages represent probability of failure, based on the percentage of structures that lie within the critical zone.

| Rock-toppling kinematic analysis | | |
|---|---|---|
| **Location in Capitólio canyon:** | **Rock-toppling site** | **Slope stability analysis site** |
| **Slope direction:** | 278/88 (post-event) | 235/85 |
| **Unweathered bedrock:** | | |
| $\phi p$= 65.2 | 19.5% | 20.1% |
| $\phi r$ = 28.0 | 19.5% | 21.9% |
| **Weathered bedrock:** | | |
| $\phi p$ = 36.6 | 19.5% | 21.9% |
| $\phi r$ = 12.8 | 19.5% | 21.9% |

"

**In the revised discussion, we highlight how, even though the stability analysis was not conducted in the site of the 2022 disaster, it can support the investigation of the factors that can lead the rock-toppling events in the canyons. This investigation is valuable in the development of disaster-prevention programs for the region. Moreover, we highlight that our results are in line with a previous study conducted in the area by Sun et al. (2024), despite some shortcomings of their analysis.**

"The analysis of the conditioning factors that control rock instabilities in the canyons, such as the structural geology and rock-mass quality, is fundamental to determining susceptibility and, hence, support these disaster-preventions strategies. The investigation of the factors that can potentially initiate slope failures gives further support to the creation of hazard and risk scenarios, which are fundamental for the implementation of contingency plans and monitoring programs. Rainfall and reservoir-level fluctuations are indicated as significant factors that can influence rock instabilities in the region, so more detailed and long-term studies on rainfall thresholds and on the impacts that reservoir water-level variations can have on slope stability, as well as that rock instabilities can have on the Furnas dam, are recommended, even though the lack of a landslide database is a great challenge.

The application of the FEM-LEM coupling in the stability analysis was fundamental to assessing the effect that rainfall and reservoir-level fluctuations had on the rock-toppling event of January 2022, both independently and combined. Rainfall is suggested as having a higher impact, although reservoir water level variation also had a role in the event's initiation and many studies have shown that landslides and rock instabilities in reservoirs can be intrinsically related to fluctuating water levels (Fujita, 1977; Hansmann et al., 2012; Meng et al., 2020). Even though our analysis is not conducted in the site that the disaster occurred, the stability assessment of a slope with a very high susceptibility to rock-toppling can provide evidences based on the FoS of the conditions that led to slope failure under a similar scenario as of that of the days leading to the 2022 event.

A recent study conducted remotely in the region by Sun et al. (2024) performed a numerical simulation of this rock-topping event, based on high-resolution data (DEM) provided by the hydropower plant company. Sun et al. (2024) also suggest that the rock-toppling event was caused by rain infiltration and weakened support at the base of the slope, although water-level fluctuations is not considered in their analysis nor a detailed structural geology analysis is performed, which, as our study suggests, are important factors that contribute to slope failures in the region. Moreover, they attribute that the erosion at the base of canyon wall is related to the occurrence of sedimentary carbonate rocks underneath the Furnas Quartzite, which, considering that we are within a complex metamorphic setting (Passos *Nappe*), is very unlikely and not observed by us during our extensive field investigations.

New References:

Fujita, H. Influence of water level fluctuations in a reservoir on slope stability. Bulletin of the International Association of Engineering Geology 16, 170–173 (1977). https://doi.org/10.1007/BF02591474

Hansmann, J., Loew, S. & Evans, K.F. Reversible rock-slope deformations caused by cyclic water-table fluctuations in mountain slopes of the Central Alps, Switzerland. Hydrogeol J 20, 73–91 (2012). https://doi.org/10.1007/s10040-011-0801-7

Meng, Q., Qian, K., Zhong, L., Gu, J., Li, Y., Fan, K., & Yan, L. (2020). Numerical analysis of slope stability under reservoir water level fluctuations using a FEM-LEM-combined method. Geofluids, 2020, Article ID 6683311. https://doi.org/10.1155/2020/6683311"

**Finally, we have also updated the introduction of the manuscript, incorporating a contextualization about the stability analysis that was conducted:**

"[…]

Different failure mechanisms can cause rock instabilities (Hoek and Bray, 1981). When the bedrock discontinuities influence slope failure, the instability can occur as a plane sliding, wedge sliding or, as in the January 2022 event, toppling (Trollope, 1969). In heavily-jointed rock slopes, the main techniques applied on the susceptibility to failure evaluation are kinematic analyses and stability assessments, which can be based on a Limit-Equilibrium Method (LEM) and/or a Finite-Element Method (FEM) (Zheng et al., 2019). A kinematic evaluation determines the potential failure modes that can occur in a jointed rock mass, based on the slope geometry, material properties and angular relations between slope face and bedrock discontinuities (Hoek and Bray, 1981). Once the failure mode is determined to be kinematically possible, the next step is the stability analysis (Norrish and Wyllie, 1996). Both the kinematic and stability analyses will determine the probability or mode of failure under a set of conditions.

In a reservoir setting, water-level fluctuations can influence slope stability and the combination of LEM and FEM methods have been demonstrated as effective in representing the effect that seepage can have on slope failure (Yin et al. 2016; Meng et al., 2020). FEM supports the estimation the dynamic seepage based on differences in groundwater/reservoir levels, while LEM is effective in representing slope stability based on FEM results based on the Factor of Safety (Yin et al. 2016; Meng et al., 2020). Fluctuations in reservoir water level can have an impact on the local groundwater level, affecting pore pressure in the discontinuities, seepage force, as well as changing rock and soil strength parameters (Meng et al., 2020).

In this context, our objective is to analyze the factors that can lead to rock instabilities in the Furnas canyons, supporting the establishment of procedures for a safer operation of the area, with the creation of a susceptibility map, so that visitors and workers are more protected and aware of the existing geohazards. Our investigation is based on extensive field campaigns and aerial surveys, which supported the structural geology analysis and rock-mass quality evaluation, as well as both the stability and kinematic analyses of the canyons. The stability analysis is conducted based on the coupling of FEM and LEM methods.

New reference
Trollope, D. H. (1969): The Stability of Rock Slopes. Vacation School in Rock Mechanics. University College of Townsville."

*2.  The triggering and worsening factors must be distinguished, even if they have the same origin.*

**Thanks for the suggestion and, with the support of the stability analysis, it was possible to more clearly distinguish the triggering (rainfall, reservoir water level variation, overland flow accumulation) from the controlling/predisposing (structural geology, erosion of the base of the canyon, canyon wall slope) factors. We have also revised the rock-toppling dynamics figure presented in the manuscript.**

**These data are shown in the revised subchapter "4.4. Rock-toppling dynamics, susceptibility mapping and safety recommendations"**

"Based on the slope stability assessment, we can interpret that the rock-toppling event in the Furnas reservoir was initiated by the combination of (i) the rapid increase in reservoir water-level, (ii) rainfall infiltration in the joint system and, potentially, the (iii) overland flow pathways that accumulate on the site of the disaster. The structural geology and geomechanical analysis, as well as the kinematic evaluation, further demonstrated that the site had a high susceptibility to toppling, especially due to the pervasive sets of unfavorably oriented joints (discontinuities) and the erosion of the base of the canyon wall. **Figure 16** summarizes the rock-toppling dynamics in the canyon, which resulted in the January 2022 disaster.

[Figure]

**Figure 16: Steps that led to the catastrophic rock-toppling event in the Capitólio canyon, with $\sigma_1$ representing the main stress tensor. 1) The intense scouring at the base of the slope, due to the waterfall flow nearby and fluctuations in the reservoir water level, undermined the underlying support of the heavily-jointed rock column. 2) Due to weathering, the sub-vertical joints widen and sub-horizontal fractures appear due to its own weight in middle and bottom portions of the rock column. 3) Water percolates and accumulates in the joints and fractures, increasing pressure on the rock mass, leading to slope failure. The increase in reservoir water level raises the local groundwater level and, consequently, the stress conditions in the rock mass, also contributing to the occurrence of rock instabilities. 4) The toppling event occurs and (5) the rock column disaggregates, due to the impact with water and the bedrock surface. Image adapted from a previous version presented in the World Landslide Forum 6 (WLF6), in Florence - Italy (2023).**

3. *The increase in water pressure is also questionable because no exceptional precipitation and a relatively low reservoir level were measured during the event. These conditions can be considered triggers for failure in highly fractured rock, but this must be argued.*

**We appreciate the suggestion. As discussed previously in this document, we have included a stability analysis based on the combination of FEM and LEM, which supported the evaluation of the effect that some previously inferred mechanisms had on rock instability. The impact of the reservoir water level was not expressive, as highlights the FoS, but it is interpreted to have contributed to the slope failure. Rainfall, however, is suggested as having a more significant role in rock instabilities in the canyon.**

4. *Additionally, the topography of the upper part of the cliff must be detailed, as it can concentrate water through overland flow, which can then enter the rock cliff.*

**Thanks for the suggestion and we have included an overland flow accumulation map, based on the digital surface model (DSM) of the region at the top of the Capitólio canyon. The map highlights how the overland flow accumulates at the site of the 2022 disaster, which can contribute to the water infiltration in the joint system of the bedrock, as well as intensify the long-term erosion/weathering on this location.**

**The methodology of the creation of the overland flow accumulation map is described below:**

"The main products that resulted from the UAV survey were the Digital Surface Model – DSM, the 3D model and the orthophotos of the four canyons. The DSM provided an accurate representation of the conditions at the top of the canyons, supporting the estimation of the predicted natural flow path over the ground, as overland flow can potentially concentrate and infiltrate in specific portions of the rock cliff, intensifying erosion and contributing to rock instabilities. The estimation of the overland flow path was conducted in GIS software, based on the "flow direction" and "flow accumulation" tools. It is important to note that the DSM was pre-processed with the hydrological tool "Fill", to remove sinks and peaks that can be resulted from data processing and potentially impact a more correct representation of the surface."

**The results of the overland flow accumulation analysis are presented in the new chapter that shows the slope stability results (4.3. Stability analysis and the factors that influence slope stability in the canyons), where we detail more quantitatively the impact of rainfall and reservoir water-level increase on slope stability, while also discussing the potential contribution of this overland flow to the rock-toppling event:**

"Furthermore, overland flow accumulation in the site of the rock-toppling event (**Figure 12**) can also contribute to increasing the water content in the fracture and joint system, adding more pressure on rock column that ultimately toppled. The overland flow pathways in the region right at the top of the rock-toppling event concentrate in the area that failed, which, besides increase water accumulation in the joints, can also intensify the erosion and increase the susceptibility of this area to rock instabilities.

[Figure]

Figure 12: Overland flow on the region at the top of the Capitólio canyon. A) Overland flow accumulation on the top of the canyon, showing that there is a significant accumulation at the location (red contour) of the rock-toppling event. B) Orthophoto of the top of the canyon, with the slope-failure location delimitated in red."

5.  *A real 3D model of the column and structure must be presented to fully understand the mode of failure, which is important for further analysis. The volume of this column has not even been estimated.*

**As highlighted previously, we did not have access to high-resolution pre-event elevation data for the site that the rock-toppling event occurred. To overcome this deficiency, we have focused our analysis in a site that is interpreted as having a similar geotechnical-geological condition as the one that the slope failure occurred. The main difference between these sites is the lack of cavities at the bottom of the slope, resulted from erosion. These cavities are interpreted as an important predisposing condition for the slope failure, contributing to the loss of underlying support of the rock mass. This is argued in the new version of the manuscript.**

**The stability analysis, as well as the updated kinematic analysis, is based on the 3D model that was acquired using UAV (Figure 13 below):**

[Figure]

Figure 13: Slope stability analysis. A) Cross-section of the location interpreted as having a similar geological-geotechnical characteristic as the site that failed on January 8, 2022. The Stereonet highlights that this portion is prone to toppling, especially considering the main tension crack in the slope. B) 3D stability analysis on Slope/W from Geostudio (Seequent, 2025), considering reservoir waterl-level fluctuations and the structural geology.

**As for the volume of the toppling body, we had presented an estimation of the volume in the first version of the manuscript, based on weight (400 ton). However, to be more precise, we chose to update it with data from the article of Sun et al. (2024), which had access to pre and post event data. This new information is presented in chapter 2, about the regional characterization of the Furnas canyons:**

"The fatal rock-toppling event occurred on January 8, 2022, at approximately 12:30h UTC, causing 10 fatalities and 32 injuries. The volume of the rock body that toppled is estimated at around $3.3 \times 10^2$ m³, according the analysis of pre and post-event elevation models that Sun et al. (2024) had access to. […]"

6. *In my opinion, using only an average stereonet for slope orientation is no longer relevant in such a case. This real 3D data must be used for the kinematic test.*

**Besides maintaining the kinematic analysis based on a general slope-face orientation, we have also conducted more kinematic tests on specific locations of the canyons, such as the site where the slope-stability**

**analysis is conducted, as well as the site that toppled on January 8, 2022. These results are presented in item 1 of this document.**

**We chose to keep the more general kinematic analysis as it supported susceptibility mapping, which was also one point suggested by the reviewer and incorporated in this new version of the manuscript. Moreover, this more specific kinematic analysis was conducted based on data from our field investigations combined with the 3-D dense point cloud data, which also supported the mapping of fracture/discontinuity density in the canyons.**

**The methodology is discussed in the revised version of the manuscript:**

"The 3D model of the canyons allowed a detailed reconstruction of the slope faces, supporting *in-situ* observations and the stability analysis. The 3D model and the 3D dense point cloud also supported the interpretation of the structural geology of the canyons, through the CloudCompareTM software (GPL Software, 2025). In the Capitólio canyon, where the rock-toppling event occurred, 500 discontinuities were measured using the "Compass" plugin, through the interpolation of planes and traces, so that the dip and dip direction of discontinuities were acquired. These 500 measurements were combined with the 557 measured *in situ* by our team.

Once the discontinuities families were identified in Stereonet, a semi-automated analysis was conducted using the "Facets" plugin for the whole canyon, as well as the other canyons in the region, based on the identified families. For the creation of the fracture-density distribution in the canyons, a shapefile with the barycentre of the fractures was used to define their spatial density in GIS software, using the "Point Density" tool. The density of fractures in each cell (0.5 m size) was calculated using a 5 m radius. As highlights Vanneschi et al. (2024), the radius choice was calibrated based on the best compromise between representativeness and detail at the work scale. Thresholds were defined to identify three density classes: low density ($\leq 2/m^2$), medium density ($5 - 8/m^2$) and high density ($> 8/m^2$), which were defined with the aid of field observations as well."

New Reference:

Vanneschi, C., Rindinella, A., & Salvini, R. (2024). Correction: Vanneschi et al. Hazard Assessment of Rocky Slopes: An Integrated Photogrammetry–GIS Approach Including Fracture Density and Probability of Failure Data. Remote Sens. 2022, 14, 1438. Remote Sensing, 16(11), 1969. https://doi.org/10.3390/rs16111969

**And the results are presented in the revised subchapter "4.1. The structural geology and rock-mass quality of the canyons":**

"Near the intersection of these perpendicular-oriented joint sets ($F_2$ and $F_3$) and fault zone (Family $F_4$), the density of fractures in the bedrock is higher, showing a stronger persistence in the canyon walls, extending from the base to the top of the rock slope (**Figure 6C**). The rock column that toppled in the event of January 2022 was located in the intersection of these structural features (**Figure 6B and 6C**), with a clear increase in discontinuities density. The fracture density map shown in **Figure 7** further highlights that in the portions that discontinuities/fractures show a higher density per square meter, the susceptibility to rock instabilities is apparently higher, with tilting

rock columns observed in these areas (**Figure 7B and 7C**), indicating movements in the rock slope. Larger apertures in the joints, especially at the top portion of these rock columns, are also observed, which are another indication of slope movement. The intersection of joint sets with perpendicular strikes favors the formation of these individual rock columns, which, combined with the horizontally dipping bedding, contribute to rock instabilities in the canyons.

[Figure]

Figure 7: Fracture density in the Capitólio canyon. A) Fracture density map, showing the regions in the rock mass with higher frequency of joints/discontinuities per square meter. B) Detail of a section in the canyon where the density of discontinuities is high. In this portion, the intersection of different joint families create rock columns, with large apertures (>10 cm), with an apparent high susceptibility to rock instabilities. C) Another section of the canyon with an apparent high susceptibility to rock instability, where the density of fractures is high."

7. *Additionally, why are overhangs not explicitly studied as seen in Figure 13b?*

**We chose not to include overhangs in the current research as it would expand the main focus of our analysis, as well as since it is not the type of slope failure that led to the disaster. We have adjusted the manuscript to focus only on areas where rock-toppling is the main geodynamic process.**

8. *A susceptibility map must also be produced, as well as an illustration of the identified structures that pose a problem for stability.*

**Thanks for the suggestion and we agree that the susceptibility map is necessary. Based on the structural geology analysis, kinematic and stability assessments, as well as our field investigations, we created a susceptibility map with four susceptibility classes for the Capitólio canyon. Below, we detail the criteria for each class, which was included in the subchapter "4.4. Rock-toppling dynamics, susceptibility mapping and safety recommendations" of the revised manuscript:**

"[…]

To support the proposal of safety recommendations for the operation of the canyons, a rock-toppling susceptibility map is proposed (**Figure 17**), based on slope angle, geomechanical and kinematic analysis, as well as our *in situ* observations and structural geology investigation. The criteria, in priority order, for the delimitation of each susceptibility class are:

- **Low susceptibility**: Slope angle lower than 70°; rock-mass quality classified as normal (structural zone C1.4); low probability (< 10%) of toppling failure according to the kinematic analysis; no movement signs observed during field investigations.

- **Moderate susceptibility**: Slope angle higher than 70°; rock-mass quality classified as normal (structural zone C1.4), low to moderate (< 20%) probability of toppling failure according to the kinematic analysis; no apparent movement signs observed during field investigations.

- **High susceptibility**: Slope angle higher than 80°; rock-mass quality classified as normal to very weak (all the structural zones), high (>20%) probability of toppling failure according to the kinematic analysis; movements signs are observed in the slopes.

- **Very high susceptibility:** Slope angle higher than 80°; rock-mass quality classified as very weak (structural zones C1.1, C1.2 and C1.3), high (>20%) probability of toppling failure according to the kinematic analysis; very apparent movement signs are observed in the slopes, with high density of fractures and joints, which also can show large apertures (> 10 cm); record of rock-toppling events.

Areas with very high susceptibility to rock instabilities (**Figure 18**), where constant monitoring should be conducted, were identified and delimitated in all four canyons. These areas are characterized by a bedrock with poor mechanical quality (C1.1, C1.2 and C1.3) and unfavorable structural geology, considering the relationship of slope face and bedrock discontinuities. Also, movement signs in the slopes can be observed in the field and are areas where the overland flow pathways tend to concentrate, which can facilitate erosion. The location of the sites that should be monitored more closely in the four canyons is available as Supplementary Information (Figure S1). The implementation of retention structures and other safety measures, such as destruction of the rock body and/or manual acceleration of the slope movement (i.e., hazard elimination practices) can potentially be implemented by public authorities and park administrators, although a more long-term analysis is needed to assess their efficacy, especially considering that it severely impacts the landscape and can impact tourist revenue. The isolation of areas that already show slope movements signs and keeping a safety distance from the canyons, especially where the rock-mass quality is weaker, can more adequately support a safer visitation experience, without visual impacts, and is strongly suggested to be implemented.

[Figure]

Figure 17: Rock-toppling susceptibility map of the Capitólio canyon.

[Figure]

Figure 18: Overview of areas with very high susceptibility to toppling. A) In the Capitólio canyon, very high susceptibility areas have a high density of fractures, with slope movement signs and fractures with large aperture (>20 cm). B) In the Cascatinha canyon, several very high susceptibility scenarios are observed along the access to the waterfall located at the end of the canyon. In this example, the underlying support of the slope is severely impacted and movement signs are observed. C) The Cabritos canyon is located near the Furnas dam. The canyon wall can reach up to 100 m high and very large rock columns show signs of movement, such as the inclination towards the canyons area and fractures with very large (>50 cm) aperture."

9. *Additionally, the term "landslide" can be replaced with "rock instability."*

**We have reviewed the terms used in the paper and replaced topple failure to rock toppling where applicable, as well as changed the term landslides to rock instabilities where suitable.**

Specific Comments

1. *Line 59: I'm not sure if this is useful. Either way, you said that tourism increased.*

**We have kept this sentence as it contributes to a better contextualization of the event and the topic of the research.**

2. *Line 66: What do you mean by "textural (2D)"?*

**Textural data refers to the lithological characteristics of the rock-mass, such as lithotype, grain size, etc. We have included some examples in the sentence to be more specific:**

"In geohazard studies that involve rock slope stability, textural (e.g., lithology, grain size) and structural (e.g., fractures, faults) data are fundamental to characterizing rock-mass structure and quality […]"

3. *Line 89: Why is the county surface area relevant to the paper?*

**We were referring to the surface area of the Furnas Reservoir in this sentence. To avoid this confusion, we chose to eliminate this information, as it is not extremely relevant.**

4. *Line 92: Provide numbers.*

**The population of Capitólio, which is around 10,000 inhabits, can reach up to 30,000 people during long weekends and holidays.**

**We have included these numbers in the manuscript:**

"[…]. The reservoir was created due to the construction of the Furnas Hydropower plant in 1958 (Godoy, 2017), and, in 2020, the tourism related to the canyons and the reservoir represented over 65% of the total Gross Domestic Product (GDP) of the surrounding municipalities, with the population tripling during holidays (Machado et al. 2020). For instance, the city of Capitólio, with approximately 10,000 inhabitants, receives up to 20,000 tourists on long weekends (Machado et al. 2020)."

5. *Line 93: What are the criteria and numbers for overtourism?*

**We have included a footnote on the word "Overtourism", with the definition based on Dodds and Butler (2019):**

"Overtourism is the excessive number of tourists at a specific destination, which can result in negative impacts on the community and natural environment (Dodds and Butler 2019)."

Reference:

Rachel Dodds, Richard Butler; The phenomena of overtourism: a review. International Journal of Tourism Cities 3 December 2019; 5 (4): 519–528. https://doi.org/10.1108/IJTC-06-2019-0090

6. *Line 108: I guess you mean paragneiss.*

**Yes, paragneiss. We have included this more specific notation of the rock type:**

"The geology is comprised of metasediments of the Araxá Group, mainly quartzite, schist and paragneiss (Heilbron et al., 2007)."

7. *Line 109: Quartzite is usually very weak due to weathering and fracturing.*

**The "strong" in this sentence refers to a well-defined sub-horizontal foliation.**

8. *Use "rock instability" instead of "landslide."*

**We have replaced the term landslide to rock instability where suitable across the whole manuscript.**

9. *Lines 145 and 152 do not have the same definition of JCS.*

**We have adjusted these inconsistencies regarding the definition of JCS, which is Joint Wall Compression Strength.**

10. *Line 159: And so what?*

**We have expanded the phrase, to be more conclusive:**

"At the time of our investigation (March 2022), the reservoir water level was at about 765 m above sea level (asl), approximately 6.1 m higher than in the day of the fatal event, which level was at 758.87 m asl on January 8."

11. *Line 187: GNSS RTK with a base?*

**We employed a GNSS PPK with base, due to communication inconsistencies between rover and the base during survey. We have expanded this information on the revised version:**

"The use of drone navigation GNSS PPK (Post-Processed Kinematic) with base (±3cm) allowed the georeferencing in a coordinate system, which was exported to a high-resolution (1.65cm/pixel) grid-based DSM and Orthomosaics. The algorithms implemented in the software are described in Verhoeven (2011). A GNSS PPK was used due to communication inconsistencies between the rover and the base during the survey."

12. *Figure 4 is too fuzzy.*

**Due to the changes in the manuscript, we chose to eliminate Figure 4 in this new version.**

*13. Lines 266–267 have no arguments.*

**We have revised the manuscript as a whole and this sentence was moved to the chapter "4.4. Rock toppling dynamics, susceptibility mapping and safety recommendations", so that it consolidates all the new analyses that were conducted in this new version, such as the stability assessment and new kinematic evaluations.**

*14. Line 272: What can be deduced if the aperture can reach 50 cm?*

**We can deduce that the rock mass has been slowly moving due to gravity. We have expanded this interpretation in the phrase:**

"Such large aperture observed in the bedrock, especially at the top portion, suggests that the rock mass has been slowly moving/toppling due to gravitational forces and weathering, which is accentuated by the heavily-jointed bedrock."

15. *Figure 6: As shown, the rainfall cannot fill the back joint. There must be a way to bring water there: overland flow or the water table level.*

**We have reviewed the figure, to include overland flow accumulation, which is interpreted to potentially have also contributed to water infiltration in the joint system of the bedrock. The revised figure, which is now Figure 16, is presented below:**

[Figure]

Figure 16: Steps that led to the catastrophic rock-toppling event in the Capitólio canyon, with $\sigma_1$ representing the main stress tensor. 1) The intense scouring at the base of the slope, due to the waterfall flow nearby and fluctuations in the reservoir water level, undermined the underlying support of the heavily-jointed rock column. 2) Due to weathering, the sub-vertical joints widen and sub-horizontal fractures appear due to its own weight in middle and bottom portions of the rock column. 3) Water percolates and accumulates in the joints and fractures, increasing pressure on the rock mass, leading to slope failure. The increase in reservoir water level raises the local groundwater level and, consequently, the stress conditions in the rock mass, also contributing to the occurrence of rock instabilities. 4) The toppling event occurs and (5) the rock column disaggregates, due to the impact with water and the bedrock surface. Image adapted from a previous version presented in the World Landslide Forum 6 (WLF6), in Florence - Italy (2023).

16. *Figure 7: Add a stereonet and illustrate the discontinuities that play a role in the picture.*

**We have revised the Figure (now Figure 6) with the suggested alterations. The revised figure is below:**

[Figure]

Figure 6: Structural analysis. A) Structural analysis of the bedrock using Stereonet. Three main joint sets were identified in the study site, as well as fault zone parallel to Family $F_3$ (n = 1077). Measurements showing Dip Direction and Dip. B) Overview of the Capitólio canyon, highlighting areas the NE-SW fault zone (Family $F_4$), which is parallel to joint family $F_3$, intersect with Family $F_2$. The rock-toppling event occurred in the far right (to the east) intersection, highlighted in the dotted yellow circle. C) The fracture density in the bedrock is intensified in areas where two perpendicular joint families intersect. The photograph is from the portion where the fatal rock-toppling event occurred, highlighting the contribution of fractures and joints to the slope failure. The Stereonet plot shows three families individualized, based on *in situ* measurements (n=56). Measurements showing Dip direction and Dip.

17. *Lines 310-312 are unclear.*

**We have removed the word "both" from the phrase, which was causing the confusion. We apply the peak friction angle in the kinematic analysis when we are evaluating a unweathered bedrock, and the residual friction angle when considering a highly weathered bedrock.**

18. *Figure 9: The same remarks apply as for Figure 7.*

**We have revised figure 9 (now Figure 15) based on the comments:**

[Figure]

**Figure 15: Details of the Capitólio canyon bedrock. A) Overview of the central area of the Capitólio canyon, showing where photos B) and C) were taken. B) A "cave" formed at the bottom of the canyon wall, due to the scouring of the rock mass due to the nearby waterfall flow and variations in the reservoir water level. C) The pervasive set of joints in the bedrock, with large aperture (up to 50 cm), is clearly seen in this portion. In this particular area, two fault zones intersect (NW-SE and NE-SW strikes), contributing to the increase in jointing and fracturing of the rock mass and facilitating slope failures. In this portion, the isolation of the area is suggested, as there is a high susceptibility to toppling. The stereonet illustrate the joint sets identified in this portion of the slope, based on 206 measurements, both *in situ* and acquired using the 3D point cloud.**

19. *Table 5: How is $\varphi_b$ calculated? Not in the formula or graphs. By the way, it is extremely high!*

**The calculation of the base friction angle is shown in Equation 1:**

$$\tau = \sigma_n \cdot tg\left[(\phi_b\ ) + JRC \cdot log_{10}\left(\frac{JCS}{\sigma_n}\right)\right] \qquad (1)$$

**In Table 5, we show the residual friction angle and peak friction angle, not the base friction angle.**

20. *Line 375: Which discontinuity sets have the largest spacing?*

**We have included the information that the Family of joint sets F2 shows the largest spacing:**

- Family $F_2$: NW-SE strike, with dip that varies from 80° to 90°, with a primary SW dip direction and, secondarily, NE dip direction (**Figure 9A**). This family shows the largest spacing between discontinuities and largest apertures, often centimetric;"

21. *Lines 401 and 413: How are these movements detected?*
22. *Line 402: This invalidates the approach of the average steronet slope orientation.*

**We have altered the sentence and included the information about the joint density map, highlighting how these slope movements are detected:**

"The fracture density map shown in **Figure 7** further highlights that in the portions that discontinuities/fractures show a higher density per square meter, the susceptibility to rock instabilities is apparently higher, with tilting rock columns observed in these areas (**Figure 7B and 7C**), indicating movements in the rock slope. Larger apertures in the joints, especially at the top portion of these rock columns, are also observed, which are another indication of slope movement. The intersection of joint sets with perpendicular strikes favors the formation of these individual rock columns, which, combined with the horizontally dipping bedding, contribute to rock instabilities in the canyons."

[Figure]

Figure 7: Fracture density in the Capitólio canyon. A) Fracture density map, showing the regions in the rock mass with higher frequency of joints/discontinuities per square meter. B) Detail of a section in the canyon where the density of discontinuities is high. In this portion, the intersection of different joint families create rock columns, with large apertures (>10 cm), with an apparent high susceptibility to rock instabilities. C) Another section of the canyon with an apparent high susceptibility to rock instability, where the density of fractures is high."

*23. Tables 8 and 9: You need to explain the percentages.*

**The percentages represent the probability of failure. We have included this information in the legends of Tables 8 and 9.**

"Table 1: Kinematic analysis results for each geomechanical compartment in the Capitólio canyon, considering rock-toppling susceptibility. $\phi r$ = Residual friction angle; $\phi p$ = Peak friction angle. The percentages represent probability of failure, based on the percentage of structures that lie within the critical zone.

| | | Rock-toppling kinematic analysis | | | | | | | | |
|---|---|---|---|---|---|---|---|---|---|---|
| | | **Geomechanical compartments** | | | | | | | | |
| | | **C1** | | **C2** | | | **C3** | | **C4** | |
| **Slope direction:** | | 229/90 | 295/90 | 330/90 | 350/90 | 180/90 | 225/90 | 315/90 | 240/90 | 195/90 |
| **Unweathered bedrock:** | $\phi p$= 65.2 | 6.4% | 7.9% | 7.4% | 11.4% | 13% | 6.9% | 7.4% | 3.9% | 8.3% |
| | $\phi r$ = 28.0 | 6.7% | 10.5% | 9.8% | 12.3% | 13.5% | 7.2% | 10.2% | 3.9% | 8.8% |
| **Weathered bedrock:** | $\phi p$ = 36.6 | 6.5% | 10.5% | 9.7% | 12.3% | 13.3% | 7.11% | 10.2% | 3.9% | 8.8% |
| | $\phi r$ = 12.8 | 6.7% | 10.5% | 9.8% | 12.3% | 14% | 7.2% | 10.2% | 3.9% | 9.3% |

Table 2: Kinematic analysis results for each geomechanical compartment in the Capitólio canyon, considering rock-toppling susceptibility. $\phi r$ = Residual friction angle; $\phi p$ = Peak friction angle. The percentages represent probability of failure, based on the percentage of structures that lie within the critical zone.

| | | Rock-toppling kinematic analysis | | | | | | | |
|---|---|---|---|---|---|---|---|---|---|
| | | **Geomechanical compartments** | | | | | | | |
| | | **C5** | **C6** | **C7** | **C8** | | **C9** | **C10** | |
| **Slope direction:** | | 142/90 | 92/90 | 54/90 | 135/90 | 95/90 | 125/90 | 80/90 | 230/90 |
| **Unweathered bedrock:** | $\phi p$= 65.2 | 19.2% | 11.6% | 24.4% | 20.4% | 11.7% | 19.3% | 13.3% | 6% |
| | $\phi r$ = 28.0 | 20.1% | 13.3% | 27% | 21.1% | 13.5% | 20.6% | 15.4% | 6.2% |
| **Weathered bedrock:** | $\phi p$ = 36.6 | 20.10% | 13.3% | 27% | 21.1% | 13.5% | 20.6% | 15.4% | 6% |
| | $\phi r$ = 12.8 | 20.4% | 13.3% | 27.2% | 21.1% | 13.5% | 20.6% | 15.6% | 6.2% |

"

*24. Figure 13: The same remarks apply as for Fig. 7. How do you get pressure with an aperture of 50–60 cm?*

**The apertures of 50-60cm are usually observed in the top section of the canyon walls, which tend to show a higher weathering degree due to overland flow and rainfall infiltration. In the bottom portion of the canyons, where the rupture surface is located, the joints and fractures show lower apertures.**

*25. Lines 473–476 are not demonstrated in the paper.*

**Based on the revision of the manuscript, with the inclusion of new analyses, such as the stability assessment and more robust structural data based on the 3D photogrammetric products, this new version demonstrates the arguments made in these lines.**

---

## Author Comment (AC2)

**Point-by-point response to comments made by Reviewer #1**

*Major suggestions:*

1. *Discuss limitations of the adapted RMR14 method, especially subjectivity in parameter scoring (e.g., discontinuity conditions, weathering). Sensitivity analysis could strengthen results.*

**In the discussion of the results, we have slightly broadened our discussion about the limitations of the adapted RMR14 method. Due to the length of this new version of the manuscript, which included the stability analysis based on FEM and LEM, as well as fracture mapping based on the UAV survey, we have chosen not perform a sensitivity analysis in the RMR14 chapter. We understand that our results are qualitative in such section, but other more quantitative analysis, such as the frequency density of fractures in the rock mass, as well as the Schmidt hammer sampling, corroborate our geomechanical assessment.**

**These changes can be observed in the revised discussions chapter:**

"**[…].** A limitation of rock-mass quality evaluation methodologies is that they can be subjective, especially without the aid of geotechnologies, and are dependent on detailed field observations (e.g., structural geology and visual analysis) that can be expensive and time consuming. Sensitivity analysis, to determine which parameter plays a more significant role on the rock-mass quality, can support decreasing subjectivity, particularly when combined with other more quantitative assessments (e.g., Schmidt Hammer) that can corroborate field interpretations."

2. *Current analysis is quite qualitative. Some simple stability analyses (e.g., limit equilibrium or FEM) to validate the inferred mechanisms (e.g., water infiltration, erosion effects) could be much better.*

**We appreciate the suggestion and we have included a stability analysis based on the combination of FEM and LEM methods, to support the evaluation of the effect that some previously inferred mechanisms had on rock instability, such as rainfall infiltration and reservoir water-level fluctuations.**
**The FEM analysis evaluated the effect of reservoir water-level fluctuations and rainfall infiltration on the rock mass, with the results being imported in our LEM analysis of the slope stability.**

**The methodology as follows:**

"Once the probabilities of the failure(s) mode(s) have been determined, the next step was the stability analysis considering the 2022 event (toppling), which compares the ratio of forces resisting failure with shear forces (i.e., the Factor of Safety – FoS) (Norrish and Wyllie, 1996). As highlighted previously, elevation data from the canyons before the event was not made available to our team despite our requests to the company that operates the hydropower plant, so the stability analysis is conducted in a portion of the Capitólio canyon (Figure 5) that is interpreted to have a similar geological-geotechnical context to the location where the rock-toppling event

occurred. While this portion did not fail on January 8, 2022, the variations in FoS can support our analysis of the conditions that led to the fatal event.

For the implementation of the FEM-LEM-combined method in the stability analysis, Seep/W (FEM) and Slope/W (LEM) are applied. Slope/W imports pore-water pressure data from Seep/W, supporting the analysis of seepage on rock instabilities in the canyon. The Method of Morgernstern-Price (M-P) is used in Slope/W, as it places no restriction on the shape of the failure surface, thus more adequate to rock slopes, satisfying both force and moment equilibrium.

As proposes Yin et al. (2016), first we consider the steady-state condition as the initial condition for the next analysis, which is when rainfall and reservoir water level fluctuations are included, both separately and combined. The Factor of Safety (FoS) in Slope/W is calculated based on the transient seepage condition resulted from Seep/W. The physical-mechanical parameters applied in the calculation are based on in situ measurements and estimations using the Schmidt Hammer, as well on literature data from Vidal et al. (2014), Lògó and Vásárhelyi et al. (2019) and Sujatono and Wijaya (2022) for Quartzites. For simplicity, strength parameters of the rock slope in natural and wet state are assumed to be the same (Tang et al., 2025).

The hydraulic conductivity (K) for a joint set of the rock mass is estimated based on the equation from Snow (1968) for fractured systems with parallel array of planar fractures (Eq. 7). The aperture of the fractures (b), and the number of fractures per unit distance of rock face (N) are required to estimate K, which are parameters estimated in our field campaigns. Other parameters are water density ($\rho$), the gravitational constant (g) and the dynamic viscosity ($\mu$).

$$K_{joint\ set} = \frac{\frac{\rho g}{\mu} N b^3}{12} \tag{7}$$

For the seepage analysis, we assume that if the elevation point is above reservoir level, the flux is zero and other far-field boundaries are assumed as non-permeable (Meng et al., 2020). The rainfall-induced seepage was simulated by applying a unit flux as no-flow boundaries.

On Seep/W, we first establish the rock-toppling area and define the rock properties, to proceed with the definition of the groundwater/reservoir head boundary. After these first two steps, the entry and exit range of the failure surface is set on Slope/W, to calculate the FoS. The slope stability is analysed in the following three situations:

- **Under a fluctuating water level condition**: As a reservoir water level fluctuates, it can impact the pore water pressure in the discontinuities, as well as potentially weaken geomechanical properties of the bedrock, facilitating erosion (Tannant et al., 2017).

- **Under rainfall conditions**: As the rainfall continues and intensifies, the infiltration of rainwater into the fractured rock mass can influence hydrostatic and hydrodynamic pressure of the slope, with the accumulation of water in the discontinuities increasing bulk density and decreasing shear resistance. Water infiltration will also impact groundwater level, reducing effective normal stress (Tannant et al., 2017).

- **Water-level fluctuations + rainfall**: Rainfall can increase bulk density and decrease shear resistance, while a fluctuating water level can impact the pore water pressure in the discontinuities, as well as the resistance to erosion of the rock mass.

[Figure]

**Figure 5: Stability analysis. A) Overview of the Capitólio canyon showing the location where the stability analysis is conducted (B) and the site where the rock-toppling event occurred (C). This picture is from before the event (February, 2019), when reservoir water level was much lower (around 755 m asl) than in February and March, 2022, when our field campaigns were conducted (765 m asl). (Source: Ion David Zarantonelli, 2019). B) Location where the analysis is conducted, interpreted as having a similar geological-geotechnical condition as the location of the rock instability. B) Post-event profile photo of the location where the rock toppling occurred."**

New references:

Lógó, B. A., Vásárhelyi, B. Estimation of the Poisson's Rate of the Intact Rock in the Function of the Rigidity. Periodica Polytechnica Civil Engineering, 63(4), pp. 1030–1037, 2019. https://doi.org/10.3311/PPci.14946

Morgenstern N R and Price V E 1965 The analysis of the stability of general slip surfaces; Geotechnique 15(1) 79–93.

Sujatono, S., Wijaya, A.E. The influence of quartz content on modulus of elasticity and Poisson's ratio in quartz sandstone. Bull Eng Geol Environ 81, 287 (2022). https://doi.org/10.1007/s10064-022-02798-6

Tang, H., Li, C., Hu, X. et al. Deformation response of the Huangtupo landslide to rainfall and the changing levels of the Three Gorges Reservoir. Bull Eng Geol Environ 74, 933–942 (2015). https://doi.org/10.1007/s10064-014-0671-z

Tannant, D. D., Giordan, D., & Morgenroth, J. (2017). Characterization and analysis of a translational rockslide on a stepped-planar slip surface. Engineering Geology, 220, 144-151. https://doi.org/10.1016/j.enggeo.2017.02.004

Vidal, FH.; Castro, NF; Azevedo, HCA. Tecnologia de rochas ornamentais: pesquisa, lavra e beneficiamento – Rio de Janeiro: CETEM/MCTI, 700p.: il. 2013.

Yin, Y., Zhang, L., Liu, Y., & Wang, H. (2016). Reservoir-induced landslides and risk control in Three Gorges project on Yangtze River, China. Journal of Rock Mechanics and Geotechnical Engineering, 8(5), 577–595. https://doi.org/10.1016/j.jrmge.2016.08.001

**From the results of the stability analysis, considering the eight-day period prior to the disaster (January 1 – 8, 2022), we could observe that the Factor of Safety (FoS) of the slope was more impacted by the combination both rainfall and reservoir water-level increase, with rainfall suggested as having a slightly more influence over the overall stability than the water-level fluctuations, when they were analyzed separately.**

**These results are described in the new subchapter "4.3. Stability analysis and the factors that influence slope stability in the canyons". This subchapter incorporated the results of the previous version of the manuscript as well, corroborating the data that was originally presented.**

"The kinematic analysis demonstrated that the canyons are susceptible to planar and toppling failure, which can be initiated by many factors, including rainfall, a fluctuating reservoir water level, excavation of the toe of the slope, among many others (Amini et al., 2012; Alejano et al., 2010; Gu and Huang, 2016; Hu et al., 2019). Even though, as highlighted by Nakamura and Wang (1990), most landslides and rock instabilities in reservoirs occur during the drawdown or impoundment, more recent studies have shown that water-level fluctuations have also the ability to significantly impact slope stability, creating a stronger water conductivity in the upper portions of the bedrock in periods of higher water level, thus affecting deformation and tensile strength (Tu et al., 2020; Li et al., 2024).

The annual variations of the Furnas water level in the last 32 years show that after a sharp decrease in 2013 (**Figure 11A**), the reservoir level has been fluctuating greatly, contrasting with the long period of somewhat stable levels between 2001 and 2012. The sharp decrease in 2013 can be associated with the drought that Southeast Brazil experienced in the 2013 – 2017 period (Finke et al., 2020), as reservoirs can be significantly affected by changes in annual and seasonal precipitation, as well as temperature variations (Mukheibir, 2013). In this period, no slope failure was reported to authorities.

[Figure]

**Figure 11: Water level of the Furnas reservoir and rainfall record. A) Long-term (32 years) variations in reservoir water level (red line) and rainfall data (blue bars). B) Reservoir water level variation in the month leading to the rock-toppling event.**

When the reservoir water level in the month leading to the slope failure is considered (**Figure 11B**), we can observe that there was a 3-m increase in this period, from 755.8 m asl on December 8, 2021, to 758.87 m asl on January 8, 2022. In the eight-day period prior to the catastrophe, the average water-level increase rate was 20 cm/day (**Table 11**). Rapid increases in water level and reservoir volume can profoundly impact the local groundwater level and stress behavior on the slope, which, as highlight by Jin et al. (2023), can cause deformation and slope failure(s).

When rainfall is considered, the accumulated precipitation reached 138 mm in the period from January 1 to January 8 (**Table 11**), with the rainfall in the 24h prior to the disaster estimated at around 35 mm. While not negligible, these values are not necessarily expressive for the region, which monthly average for summer months is around 280 mm. However, as highlighted previously, the pluviometer network in the region is very sparse, so the actual rainfall at the Capitólio canyon could be higher as the rain gauge that recorded the rain event is located more than 10 km away from the site of the event. Moreover, the effect of water infiltration in fractured rock masses is progressive with time (Yunjin et al., 2001; Liu et al. 2025), as the flow depends on the connectivity of the joints and fractures. And, when we analyze the rainfall in the month leading to the slope failure (ca. 355 mm), which was concentrated mainly in the period between December 28, 2021, and January 05, 2022 (**Figure 11B**), the infiltration can potentially have had significant role in the rock-toppling event initiation.

**Table 11: Evolution of the Factor of Safety (FoS) considering the effect of rainfall, reservoir water level fluctuations and the combination of both. The rock-toppling event occurred on January 8, 2022. The stability analysis considers a site in the canyon that is interpreted to have a similar geological-geotechnical condition as the one where the slope failure occurred, since no pre-event data was made available to our team.**

| Date (dd/mm/yyyy) | Rainfall (mm) | Reservoir water level on the days leading to the rock-toppling event (m asl) | Reservoir water level assumed in the stability analysis (m asl) | Factor of Safety analysis | | |
|---|---|---|---|---|---|---|
| | | | | FoS Reservoir level | FoS Rainfall | FoS Rain+reservoir |
| 01/01/2022 | 21 | 757.48 | 766.16 | 1.309 | 1.309 | 1.309 |
| 02/01/2022 | 24 | 757.64 | 766.32 | 1.305 | 1.306 | 1.305 |
| 03/01/2022 | 33 | 757.81 | 766.49 | 1.302 | 1.302 | 1.302 |
| 04/01/2022 | 5 | 758.01 | 766.69 | 1.302 | 1.3 | 1.299 |
| 05/01/2022 | 18 | 758.21 | 766.89 | 1.301 | 1.298 | 1.297 |
| 06/01/2022 | 2 | 758.39 | 767.07 | 1.301 | 1.298 | 1.297 |
| 07/01/2022 | 27 | 758.64 | 767.32 | 1.299 | 1.297 | 1.294 |
| 08/01/2022 | 8 | 758.87 | 767.55 | 1.299 | 1.297 | 1.293 |

Furthermore, overland flow accumulation in the site of the rock-toppling event (**Figure 12**) can also contribute to increasing the water content in the fracture and joint system, adding more pressure on rock column that ultimately toppled. The overland flow pathways in the region right at the top of the rock-toppling event concentrate in the area that failed, which, besides increase water accumulation in the joints, can also intensify the erosion and increase the susceptibility of this area to rock instabilities.

[Figure]

**Figure 12: Overland flow on the region at the top of the Capitólio canyon. A) Overland flow accumulation on the top of the canyon, showing that there is a significant accumulation at the location (red contour) of the rock-toppling event. B) Orthophoto of the top of the canyon, with the slope-failure location delimitated in red.**

Thus, to quantitatively analyze the influence of the reservoir water-level fluctuations and rainfall on the slope stability, we employed Seep/W considering the period of eight days prior to the rock-toppling event, from January 1 to January 8. First, the dynamic seepage field is calculated based on FEM with the varying water level and rainfall data. The partitions of the material are then reassigned at every step with the dynamic seepage field, so that the FoS can be finally computed by the Morgernstein-Price Method on Slope/W (LEM) (**Figure 13**). The physical-mechanical parameters (**Table 12**) applied in the stability analysis are based on field surveys and geomechanical investigation of the rock mass (shear resistance parameters and hydraulic conductivity), while the Poisson's ratio and Elastic Modulus are based on average data for Quartzites (Sujatono and Wijaya, 2022; Lógó and Vásárhelyi, 2019).

The stability analysis is conducted in a site in the Capitólio canyon that is similar to the one where the rock-toppling occurred (**Figure 5A** and **Figure 13A**), as no high-resolution data from before the 2022 event was made available to our team. As in the date of our photogrammetric data acquisition (May 2022) the reservoir water level was different from the day of the event (January 2022), we performed our seepage analysis using FEM assuming a daily increase similar to what occurred in the days leading to the disaster (**Table 11** and **Figure 13B**).

**Table 12: Physical-mechanical parameters applied in the rock slope stability analysis.**

| Material | Unit Weight (kN/m³) | Elastic modulus (Gpa) | Poisson's ratio | Shear Strength | | Hydraulic conductivity (m/s) |
| --- | --- | --- | --- | --- | --- | --- |
| | | | | Cohesion (kPa) | Internal Friction Angle (°) | |
| Bedrock (Furnas Quartzite) | 25 | 20 | 0.18 | 2000 | 28 | $6\times10^{-4}$ |

[Figure]

**Figure 13: Slope stability analysis. A) Cross-section of the location interpreted as having a similar geological-geotechnical characteristic as the site that failed on January 8, 2022. The Stereonet highlights that this portion is prone to toppling, especially considering the main tension crack in the slope. B) 3D stability analysis on Slope/W from Geostudio (Seequent, 2025), considering reservoir waterl-level fluctuations and the structural geology.**

When only water-level fluctuations were considered, the numerical results show that it impacted slightly the FoS of the slope, decreasing from 1.309 to 1.299 in the eight-day period (**Figure 14 and Table 11**). The impact of rainfall on the FoS was slightly higher, decreasing to 1.297 from 1.309 in the considered timeline. The combined effect of rainfall and reservoir water-level fluctuation impacted the FoS the most, decreasing from 1.309 to 1.293. While these scenarios were not sufficient to cause a slope failure at this location, it can suggest how these different factors impacted the site that did fail on January 2022. Thus, based on this relationship between FoS, water-level variations and precipitation, it can be suggested that both rainfall and reservoir water level had a role on the rock-toppling initiation, with rainfall infiltration with a slightly stronger influence on slope stability.

Furthermore, during our field surveys, it was observed the formation of cavities at the bottom of the slope where rock-toppling event occurred, caused by the long-term erosion in this region (**Figure 15A and 15B**). These cavities are interpreted to be one of the main predisposing factors to slope failure and are potentially resulted from the effect of waves in the reservoir, caused by wind and boat circulation, as well as by water flow from the nearby waterfall (**Figure 15A**). Moreover, the long-term fluctuations in the reservoir water level (**Figure 11A**) are also inferred as one of the contributors to the formation of these cavities, weakening the geomechanical properties of the bedrock (Tannant et al., 2017). These cavities caused the middle section to lose underlying support, with an added pressure of water in the joints. As pointed by Cruden (1991), self-weight alone is not sufficient to cause

toppling, with external forces needed to initiate the process. In the site that we performed the stability analysis, these cavities were not observed (**Figure 15C**), which can contribute to increasing the stability of that portion of the canyon.

[Figure]

**Figure 14: Evolution of Factor of Safety based on the effect of rainfall, reservoir level increase and the combination of both processes.**

[Figure]

**Figure 15: Details of the Capitólio canyon bedrock. A) Overview of the central area of the Capitólio canyon, showing where photos B) and C) were taken. B) A "cave" formed at the bottom of the canyon wall, due to the scouring of the rock mass due to the nearby waterfall flow and variations in the reservoir water level. C) The pervasive set of joints in the bedrock, with large aperture (up to 50 cm), is clearly seen in this portion. In this particular area, two fault zones intersect (NW-SE and NE-SW strikes), contributing to the increase in jointing and fracturing of the rock mass and facilitating slope failures. In this portion, the isolation of the area is suggested, as there is a high susceptibility to toppling. The stereonet illustrate the joint sets identified in this portion of the slope, based on 206 measurements, both *in situ* and acquired using the 3D point cloud.**

New references:

Alejano, L. R., Ferrero, A. M., Ramírez-Oyanguren, P., & Álvarez-Fernández, M. I. (2011). Comparison of limit-equilibrium, numerical and physical models of wall slope stability. International Journal of Rock Mechanics & Mining Sciences, 48(1), 16–26. https://doi.org/10.1016/j.ijrmms.2010.06.013

Amini A., Melville B. W., Ali T. M., Ghazali A. H. (2012). Clear-Water Local Scour Around Pile Groups in Shallow-Water Flow. J. Hydraulic Eng. 138 (2), 177–185. doi: 10.1061/(asce)hy.1943-7900.0000488

Gu, D., & Huang, D. (2016). A complex rock topple-rock slide failure of an anaclinal rock slope in the Wu Gorge, Yangtze River, China. Engineering Geology, 208, 165–180. https://doi.org/10.1016/j.enggeo.2016.04.037

Hu, X., He, C., Zhou, C., Xu, C., Zhang, H., Wang, Q., & Wu, S. (2019). Model test and numerical analysis on the deformation and stability of a landslide subjected to reservoir filling. Geofluids, Article ID 5924580, 1–15. https://doi.org/10.1155/2019/5924580

Liu, X., Sun, J., Liu, B., Kang, Y., Tian, Y., & Zhou, Y. (2025). Grouting flow in deep fractured rock: A state-of-the-art review of theory and practice. Fluid Dynamics & Materials Processing, 21(8), 2047–2073. https://doi.org/10.32604/fdmp.2025.068268 techscience.com

Yunjin, Hu, Baoyu, Su, and Zhan Melli. "Simulation of Water Flow In Fractured Rock Mass Due to Surface Infiltration And Engineering Application." Paper presented at the ISRM International Symposium - 2nd Asian Rock Mechanics Symposium, Beijing, China, September 2001.

**In the discussion, we highlight how, even though the stability analysis was not conducted in the site of the 2022 disaster, it can support the investigation of the factors that can lead the rock-toppling events in the canyons. This investigation is valuable in the development of disaster-prevention programs for the region. Moreover, we highlight that our results are in line with a previous study conducted in the area by Sun et al. (2024), despite some shortcomings of their analysis:**

"The analysis of the conditioning factors that control rock instabilities in the canyons, such as the structural geology and rock-mass quality, is fundamental to determining susceptibility and, hence, support these disaster-preventions strategies. The investigation of the factors that can potentially initiate slope failures gives further support to the creation of hazard and risk scenarios, which are fundamental for the implementation of contingency plans and monitoring programs. Rainfall and reservoir-level fluctuations are indicated as significant factors that can influence rock instabilities in the region, so more detailed and long-term studies on rainfall thresholds and on the impacts that reservoir water-level variations can have on slope stability, as well as that rock instabilities can have on the Furnas dam, are recommended, even though the lack of a landslide database is a great challenge.

The application of the FEM-LEM coupling in the stability analysis was fundamental to assessing the effect that rainfall and reservoir-level fluctuations had on the rock-toppling event of January 2022, both independently and combined. Rainfall is suggested as having a higher impact, although reservoir water level variation also had a role in the event's initiation and many studies have shown that landslides and rock instabilities in reservoirs can be intrinsically related to fluctuating water levels (Fujita, 1977; Hansmann et al., 2012; Meng et al., 2020). Even though our analysis is not conducted in the site that the disaster occurred, the stability assessment of a slope with a very high susceptibility to rock-toppling can provide evidences based on the FoS of the conditions that led to slope failure under a similar scenario as of that of the days leading to the 2022 event.

A recent study conducted remotely in the region by Sun et al. (2024) performed a numerical simulation of this rock-topping event, based on high-resolution data (DEM) provided by the hydropower plant company. Sun et al. (2024) also suggest that the rock-toppling event was caused by rain infiltration and weakened support at the base

of the slope, although water-level fluctuations is not considered in their analysis nor a detailed structural geology analysis is performed, which, as our study suggests, are important factors that contribute to slope failures in the region. Moreover, they attribute that the erosion at the base of canyon wall is related to the occurrence of sedimentary carbonate rocks underneath the Furnas Quartzite, which, considering that we are within a complex metamorphic setting (Passos *Nappe*), is very unlikely and not observed by us during our extensive field investigations."

**Finally, we have also updated the introduction of the manuscript, incorporating a contextualization about the stability analysis conducted:**

"[…]

Different failure mechanisms can cause rock instabilities (Hoek and Bray, 1981). When the bedrock discontinuities influence slope failure, the instability can occur as a plane sliding, wedge sliding or, as in the January 2022 event, toppling (Trollop, 1969). In heavily-jointed rock slopes, the main techniques applied on the susceptibility to failure evaluation are kinematic analyses and stability assessments, which can be based on a Limit-Equilibrium Method (LEM) and/or a Finite-Element Method (FEM) (Zheng et al., 2019). A kinematic evaluation determines the potential failure modes that can occur in a jointed rock mass, based on the slope geometry, material properties and angular relations between slope face and bedrock discontinuities (Hoek and Bray, 1981). Once the failure mode is determined to be kinematically possible, the next step is the stability analysis (Norrish and Wyllie, 1996). Both the kinematic and stability analyses will determine the probability or mode of failure under a set of conditions.

In a reservoir setting, water-level fluctuations can influence slope stability and the combination of LEM and FEM methods have been demonstrated as effective in representing the effect that seepage can have on slope failure (Yin et al. 2016; Meng et al., 2019). FEM supports the estimation the dynamic seepage based on differences in groundwater/reservoir levels, while LEM is effective in representing slope stability based on FEM results based on the Factor of Safety (Yin et al. 2016; Meng et al., 2019). Fluctuations in reservoir water level can have an impact on the local groundwater level, affecting pore pressure in the discontinuities, seepage force, as well as changing rock and soil strength parameters (Meng et al., 2019).

In this context, our objective is to analyze the factors that can lead to rock instabilities in the Furnas canyons, supporting the establishment of procedures for a safer operation of the area, with the creation of a susceptibility map, so that visitors and workers are more protected and aware of the existing geohazards. Our investigation is based on extensive field campaigns and aerial surveys, which supported the structural geology analysis and rock-mass quality evaluation, as well as both the stability and kinematic analyses of the canyons. The stability analysis is conducted based on the coupling of FEM and LEM methods."

> 3. *Clarify why toppling susceptibility is low overall yet critical in specific zones (e.g., fault intersections). Contrast with field evidence more explicitly.*

**Probably due to the generalization of the slope-face direction, specific zones can show a higher susceptibility to toppling than the overall susceptibility of the compartment. Compartment C2 is a good example, as the**

kinematic analysis of a specific portion of this canyon wall shows a higher susceptibility than a more generalized slope-face direction. This section, which is also where the rock-toppling stability analysis is performed, shows a similar susceptibility to that of the site of the January 2022 event. We have clarified these points in the text and added the kinematic analysis of both this specific portion in compartment C2 and of the site of the rock-toppling event to support our interpretations:

"Geomechanical compartments with slopes faces with dip direction/dip of 142/90 (compartment C5), 135/90 (C7), 125/90 (C8) and 54/90 (C6) tend to show a higher susceptibility to toppling when compared to other slope faces (**Table 9**). These compartments are notable due to the formation of rock columns (**Figure 7C**), up to 50 m high, and are located in sections of the canyons that can show movement signs, such as joints with larger apertures (>10 cm) and tilted rock columns. Moreover, these areas and compartments are also associated to a higher frequency of joints/discontinuities in the rock mass, as highlights the fracture density map (**Figure 7**).

Although not showing an overall high susceptibility to toppling in the kinematic analysis, probably due to the generalization of the slope-face direction, Compartment C2 shows a very high joint frequency (**Figure 5B** and **Figure 7A**) and signs of slope movements, with a tilting 20 m rock column that can potentially experience a similar fate to the slope that toppled on January 2022. The kinematic analysis of this section in Compartment C2, which is interpreted to have a similar condition to the site where the 2022 rock-toppling event occurred, further highlights that this specific region has a high susceptibility to toppling, with a similar probability to this geodynamic process as the slope failure site (**Table 10** and **Figure 10**).

**Table 1: Kinematic analysis results for each geomechanical compartment in the Capitólio canyon, considering rock-toppling susceptibility. $\phi r$ = Residual friction angle; $\phi p$ = Peak friction angle. The percentages represent probability of failure, based on the percentage of structures that lie within the critical zone.**

| | | Rock-toppling kinematic analysis | | | | | | | | |
|---|---|---|---|---|---|---|---|---|---|---|
| | | Geomechanical compartments | | | | | | | | |
| | | C1 | | C2 | | | C3 | | C4 | |
| Slope direction: | | 229/90 | 295/90 | 330/90 | 350/90 | 180/90 | 225/90 | 315/90 | 240/90 | 195/90 |
| Unweathered bedrock: | $\phi p$= 65.2 | 6.4% | 7.9% | 7.4% | 11.4% | 13% | 6.9% | 7.4% | 3.9% | 8.3% |
| | $\phi r$ = 28.0 | 6.7% | 10.5% | 9.8% | 12.3% | 13.5% | 7.2% | 10.2% | 3.9% | 8.8% |
| Weathered bedrock: | $\phi p$ = 36.6 | 6.5% | 10.5% | 9.7% | 12.3% | 13.3% | 7.11% | 10.2% | 3.9% | 8.8% |
| | $\phi r$ = 12.8 | 6.7% | 10.5% | 9.8% | 12.3% | 14% | 7.2% | 10.2% | 3.9% | 9.3% |

**Table 2: Kinematic analysis results for each geomechanical compartment in the Capitólio canyon, considering rock-toppling susceptibility. $\phi r$ = Residual friction angle; $\phi p$ = Peak friction angle. The percentages represent probability of failure, based on the percentage of structures that lie within the critical zone.**

| | | Rock-toppling kinematic analysis | | | | | | | |
|---|---|---|---|---|---|---|---|---|---|
| | | Geomechanical compartments | | | | | | | |
| | | C5 | C6 | C7 | C8 | | C9 | C10 | |
| Slope direction: | | 142/90 | 92/90 | 54/90 | 135/90 | 95/90 | 125/90 | 80/90 | 230/90 |
| Unweathered bedrock: | $\phi p$= 65.2 | 19.2% | 11.6% | 24.4% | 20.4% | 11.7% | 19.3% | 13.3% | 6% |
| | $\phi r$ = 28.0 | 20.1% | 13.3% | 27% | 21.1% | 13.5% | 20.6% | 15.4% | 6.2% |

| Weathered bedrock: | $\phi p = 36.6$ | 20.10% | 13.3% | 27% | 21.1% | 13.5% | 20.6% | 15.4% | 6% |
| | $\phi r = 12.8$ | 20.4% | 13.3% | 27.2% | 21.1% | 13.5% | 20.6% | 15.6% | 6.2% |

[Figure]

**Figure 10: Kinematic analysis of the rock-toppling site (left) and the site that is interpreted as having a similar geological-geotechnical condition as the one that failed on January 2022 (right). The kinematic analysis was conducted considering weathered and unweathered portions of the rock mass, as well as both the peak and residual friction angle.**

**Table 10: Kinematic analysis results for the rock toppling site and the site that is interpreted as having a similar geological-geotechnical condition as the site that failed on January 2022. $\phi r$ = Residual friction angle; $\phi p$ = Peak friction angle. The percentages represent probability of failure, based on the percentage of structures that lie within the critical zone.**

| Rock-toppling kinematic analysis | | |
|---|---|---|
| **Location in Capitólio canyon:** | **Rock-toppling site** | **Slope stability analysis site** |
| **Slope direction:** | 278/88 (post-event) | 235/85 |
| **Unweathered bedrock:** $\phi p= 65.2$ | 19.5% | 20.1% |
| $\phi r = 28.0$ | 19.5% | 21.9% |
| **Weathered bedrock:** $\phi p = 36.6$ | 19.5% | 21.9% |
| $\phi r = 12.8$ | 19.5% | 21.9% |

4. *Some quantitative evidences (e.g., pore pressure modeling) to support the hypothesis that rapid water-level rise contributed to the 2022 event would be encouraged. May be the results come from reservoir-induced landslides (e.g., Three Gorges) are useful.*

**For the analysis of the pore-pressure changes in the fractured rock mass, which could have had contributed to the rock-toppling event, we have combined a FEM and LEM analysis, as discussed previously in this document. The FEM analysis, conducted in Seep/W, was applied in the analysis of the impact that the increasing reservoir water level and rainfall infiltration had on the rock-mass strength, with the pore-pressure results being imported in Slope/W, to assess the rock slope stability. We would like to express again our appreciation for the suggestion to conduct such analysis in the improvement of our manuscript.**

5. Please expand on how fault intersections localize instability and include fracture density maps or statistical analysis of joint spacing.

**We have included a fracture-density map to highlight that in the portions where the fracture density is higher, we can observe slope movement signs, such as the formation of tilting rock columns and a larger aperture in the joints. These areas with higher fracture density are interpreted to be associated to the intersection of the NW-SE and NE-SW structural features, based on our structural geology analysis and the regional geological literature.**

**We have presented the methodology of the fracture density map in the revised methodology chapter (3.2 - Photogrammetric data acquisition, processing and related products):**

"Furthermore, the 3D model of the canyons allowed a detailed reconstruction of the slope faces, supporting *in-situ* observations and the stability analysis. The 3D model and the 3D dense point cloud also supported the interpretation of the structural geology of the canyons, through the CloudCompareTM software (GPL Software, 2025). In the Capitólio canyon, where the rock-toppling event occurred, 500 discontinuities were measured using the "Compass" plugin, through the interpolation of planes and traces, so that the dip and dip direction of discontinuities were acquired. These 500 measurements were combined with the 557 measured *in situ* by our team.

Once the discontinuities families were identified in Stereonet, a semi-automated analysis was conducted using the "Facets" plugin for the whole canyon, as well as the other canyons in the region, based on the identified families. For the creation of the fracture-density distribution in the canyons, a shapefile with the barycentre of the fractures was used to define their spatial density in GIS software, using the "Point Density" tool. The density of fractures in each cell (0.5 m size) was calculated using a 5 m radius. As highlights Vanneschi et al. (2024), the radius choice was calibrated based on the best compromise between representativeness and detail at the work scale. Thresholds were defined to identify three density classes: low density ($\leq 2/m^2$), medium density ($5 - 8/m^2$) and high density ($> 8/m^2$), which were defined with the aid of field observations as well."

**And the results are presented in the revised subchapter "4.1. The structural geology and rock-mass quality of the canyons":**

"Near the intersection of these perpendicular-oriented joint sets ($F_2$ and $F_3$) and fault zone (Family $F_4$), the density of fractures in the bedrock is higher, showing a stronger persistence in the canyon walls, extending from the base to the top of the rock slope (**Figure 6C**). The rock column that toppled in the event of January 2022 was located in the intersection of these structural features (**Figure 6B and 6C**), with a clear increase in discontinuities density. The fracture density map shown in **Figure 7** further highlights that in the portions that discontinuities/fractures show a higher density per square meter, the susceptibility to rock instabilities is apparently higher, with tilting rock columns observed in these areas (**Figure 7B and 7C**), indicating movements in the rock slope. Larger apertures in the joints, especially at the top portion of these rock columns, are also observed, which are another indication of slope movement. The intersection of joint sets with perpendicular strikes favors the formation of these individual rock columns, which, combined with the horizontally dipping bedding, contribute to rock instabilities in the canyons.

[Figure]

**Figure 7: Fracture density in the Capitólio canyon. A) Fracture density map, showing the regions in the rock mass with higher frequency of joints/discontinuities per square meter. B) Detail of a section in the canyon where the density of discontinuities is high. In this portion, the intersection of different joint families create rock columns, with large apertures (>10 cm), with an apparent high susceptibility to rock instabilities. C) Another section of the canyon with an apparent high susceptibility to rock instability, where the density of fractures is high.**
"

6.  *Some other policy and risk managements are suggested to discuss, e.g., real-time monitoring, exclusion zones, conflicts with tourism revenue and so on.*

**Thank you for the suggestion and we have expanded the discussion to mention of the suggested topics, which are very important.**

**We have updated the paragraph where we discuss some of the proposals for safety measures to avoid future disaster (subchapter 4.4. Rock-toppling dynamics, susceptibility mapping and safety recommendations):**

"[…]

The implementation of retention structures and other safety measures, such as destruction of the rock body and/or manual acceleration of the slope movement (i.e., hazard elimination practices) can potentially be implemented by public authorities and park administrators, although a more long-term analysis is needed to assess their efficacy, especially considering that it severely impacts the landscape and can impact tourist revenue. The isolation of areas that already show slope movements signs and keeping a safety distance from the canyons, especially where the rock-mass quality is weaker, can more adequately support a safer visitation experience, without visual impacts, and is strongly suggested to be implemented."

**As well as in our discussion, where we discuss some real-time monitoring techniques and why some more simple measures are potentially more useful, considering the political and economic interest in the topic by local stakeholders:**

"Our study is resulted from several months of field investigations and data analysis, as well as years (approx. 2 years) of discussions with local stakeholders and workers about the feasibility of a particular susceptibility assessment methodology that can support hazard and risk-reduction strategies. Rockfall and rock-toppling real-time monitoring programs based on the detection of surface deformation, such as InSAR (Interferometry Synthetic Aperture Radar) technologies, have been implemented in other parts of the world (e.g., Norway, Switzerland) and are demonstrated to be effected in preventing disasters related to rock instabilities (Matteo et al., 2017; Carlà et al., 2019; Sarro et al., 2025). However, taking into consideration the resources that local stakeholders are willing to invest in disaster-prevention programs, the restriction of tourist access to certain areas, as well as maintaining a safety distance from the slopes in the canyons, are strategies that are more feasible to be implemented, while also having a higher probability of being kept in place for a longer period of time despite changes in local governance. These preventive measures were suggested by the authors to the local authorities and are currently in place (2025)."

Minor suggestions:

1. *The abstract can be improved. For example, the connection between geotourism growth and landslide risks in Brazil by citing specific statistics about tourism-dependent economies in mountainous regions.*

**We have updated the abstract based on the suggestion. The new abstract can be found below:**

**Abstract.** In Geotourism-dependent regions in Brazil, landslide-related disasters can significantly impact their socioeconomic foundations. Geotourism has been on the rise in the country, although restoring tourist confidence in such regions can be a challenge when a disaster related to a natural phenomenon occurs. An example is the rock-toppling event that occurred on January 2022 in one of the four canyons located in the Furnas reservoir in Southeast Brazil, which caused 10 fatalities. Visitation fear and their temporary closure severely impacted the economy of the surrounding municipalities that rely on tourism. To support a safer operation of the canyons to visitation, our study investigates the factors that can lead to rock instabilities in the canyons, with the creation of a susceptibility map, based on the combination of field investigations, rock-mass quality evaluation (RMR14) and both kinematic and stability analyses. The stability analysis, based on the combination of FEM and LEM methods, indicates that the rock-toppling event was initiated by a combination of different factors, such as rainfall infiltration and overland-flow accumulation in the unfavorably-oriented joints of the bedrock, as well as by the influence of reservoir water level fluctuations, with the rapid increase in the days prior to the catastrophe potentially altering pore pressure in the discontinuities. Moreover, the long-term erosion at the base of the slope caused by the nearby waterfall flow, waves and water-level fluctuations, weakened rock-mass support and contributed to the rock-toppling event. The structural geology and the kinematic analysis demonstrated that the canyons are prone to slope failures, with specific locations in the slopes showing a higher rock-toppling susceptibility, especially where two perpendicularly-oriented structural features (NW-SE and NE-SW strikes) intersect. The RMR14 method adapted to open rock slopes further supported the estimation of the bedrock's geomechanical properties, identifying structural zones in the rock mass that are more prone to slope failure(s) and, as a consequence, should be monitored. These analyses and field observations supported the creation of a susceptibility map, supporting the establishment of visitation procedures in the canyons, so that tourists and workers are more protected and aware of the existing geohazards.

**Keywords:** Landslides, Structural geology, Rock-mass quality, Stability analysis, Natural hazards.

2. *The exact advantages of the proposed RMR14 method.*

**The proposed RMR14 method has the advantage of being an "update" of the traditional RMR method (Bieniawski, 1989), reducing some subjectivities that were typical of the 1989 version, such as the replacement of the water effect with the intact rock alterability, which can be estimated from Schmidt hammer data. We discuss this advantage in the discussion chapter:**

"Furthermore, the application of the adapted RMR14 is an easy to comprehend and replicate method that can help to determine rock-instabilities susceptibility, which can be applied in other tourist areas of the Furnas reservoir,

such as waterfalls and rock slopes used for climbing. An advantage of this methodology is that it is based on an update of the original RMR method and reduces some of the more subjective interpretations, such as the effect the water has on the slopes, replacing it with parameters that can be estimated more objectively using the Schmidt hammer. […]"

3. The term should be consistent cross the whole paper (e.g., rock toppling vs. topple failure).

**We have reviewed the terms used in the paper and replaced topple failure to rock toppling where applicable, as well as changed the term landslides to rock instabilities where suitable.**

4. The figure quality should be improved.

**We have reviewed all the figures to adjust quality and representativeness.**